

# Water, Energy, and Carbon with Artificial Neural Networks (WECANN): A statistically-based estimate of global surface turbulent fluxes using solar-induced fluorescence

Seyed Hamed Alemohammad[1,2], Bin Fang[1,2], Alexandra G. Konings[3], Julia K. Green[1,2], Jana Kolassa[4,5], Catherine Prigent[1,6], Filipe Aires[1,6], Diego Miralles[7,8], Pierre Gentine[1,2,9]

[1]Department of Earth and Environmental Engineering, Columbia University, New York, 10027, USA
[2]Columbia Water Center, Columbia University, New York, 10027, USA
[3]Department of Earth System Science, Stanford University, Stanford, 94305, USA
[4]Universities Space Research Association/NPP, Columbia, MD, 21046, USA
[5]Global Modeling and Assimilation Office, NASA Goddard Spaceflight Center, Greenbelt, MD, 20771, USA
[6]Observatoire de Paris, Paris, 75014, France
[7]Department of Earth Sciences, VU University Amsterdam, Amsterdam, 1081HV, The Netherlands
[8]Laboratory of Hydrology and Water Management, Ghent University, Ghent, B-9000, Belgium
[9]Earth Institute, Columbia University, New York, 10027, USA

*Correspondence to*: Seyed Hamed Alemohammad (sha2128@columbia.edu)

**Abstract.** A new global estimate of surface turbulent fluxes, including latent heat flux (LE), sensible heat flux (H), and gross primary production (GPP) is developed using remotely sensed Solar-Induced Fluorescence (SIF) and other radiative and meteorological variables. The approach uses an artificial neural network (ANN) with a Bayesian perspective to learn from the training datasets: a target input dataset is generated using three independent data sources and a triple collocation (TC) algorithm to define a prior distribution. The new retrieval, named Water, Energy, and Carbon with Artificial Neural Networks (WECANN), provides surface turbulent fluxes from 2007 to 2015 at $1° \times 1°$ spatial resolution and on monthly time resolution. The quality of ANN training is assessed using the target data, and the WECANN retrievals are validated using FLUXNET tower measurements across various climates and conditions. WECANN performs well in most cases and is strongly constrained by SIF information. The impact of SIF on WECANN retrievals is evaluated by removing it from the input dataset of the ANN, and it shows that SIF has significant influence, especially in regions of high vegetation cover and in humid conditions. When compared to *in situ* eddy covariance observations, WECANN typically outperforms other estimates, particularly for sensible and latent heat fluxes.

## 1 Introduction

Turbulent fluxes from the land surface to the atmosphere, particularly sensible heat flux (H), latent heat flux (LE), gross primary production (GPP) and net primary production (NPP) are key to understanding ecosystem response to climate and the feedback on the overlying atmosphere, as well as constraining the global carbon, water and energy cycles. In recent years, there has been substantial effort towards estimating these surface fluxes from remote sensing observations at a global scale (see e.g. Fisher et al., 2008; Jiang and Ryu, 2016; Jiménez et al., 2009, 2011; Jung et al., 2009; Miralles et al., 2011a; Mu et al., 2007; Mueller et al., 2011). Two different approaches have been used to estimate these surface fluxes from remote sensing information. The first approach uses physically-based or semi-empirical models (e.g. the Priestley-Taylor or Penman-Monteith equations in the case of ET, or a light use efficiency model in the case of GPP) informed by remote sensing information (e.g. vegetation indices, infrared temperature, microwave soil moisture), often in combination with reanalysis meteorological forcing data (Fisher et al., 2008; Miralles et al., 2011a; Mu et al., 2007; Zhang et al., 2016b; Zhao et al., 2005; Zhao and Running, 2010). These approaches are sensitive to the assumptions and imperfections of the underlying flux models. The second approach, employed by the Max Planck



Institute for Biogeochemistry model (MPI-BGC) uses machine learning (e.g. a model tree ensemble) to determine fluxes (LE, H, and GPP) from meteorological drivers and optical remote sensing data. Like all supervised machine learning models, the MPI-BGC method relies on a training dataset to determine the non-linear statistical relationships. In this case, *in situ* turbulent flux measurements from eddy-covariance towers are used (Beer et al., 2010; Jung et al., 2011). Such an approach relies implicitly on

an assumption that a long temporal record of fluxes at a small number of sites captures the full range of behavior and sensitivities of terrestrial ecosystems around the globe. In addition, extreme and therefore rare events may be difficult to capture based on the limited data availability.

Alternatively, one can use a machine learning Artificial Neural Network (ANN) approach trained on globally-representative but noisy estimates of the fluxes (such as those from models) to parameterize the non-linear statistical relationships between remote

sensing observations and surface fluxes. This approach has been successfully used for global soil moisture retrieval (Aires et al., 2012; Kolassa et al., 2013, 2016; Rodríguez-Fernández et al., 2015) and surface heat flux retrieval (Jiménez et al., 2009). Such ANNs require a target dataset for training. Climate model simulations of the relevant geophysical variable are usually used as the training dataset to facilitate the assimilation of retrievals into the model (Aires et al., 2012; Kolassa et al., 2013, 2016). However, the resulting fluxes estimated by the ANN often show some of the same biases as the simulations used to train the network

(Rodríguez-Fernández et al., 2015), even if improvements can be achieved such as a more realistic seasonal cycle (Jiménez et al., 2009).

In this study, we develop an ANN approach to retrieve monthly surface fluxes at the global scale. The network uses remotely sensed solar-induced fluorescence (SIF) estimates in addition to other data including precipitation, temperature, soil moisture, snow cover, and net radiation as inputs (predictor). To reduce biases, we introduce a Bayesian perspective to generate the training dataset for the

ANN. Multiple estimates of each of the fluxes are selected according to a prior probability that reflects the quality and information content of the dataset at the particular pixel of interest (details are provided in Section 3.2). This approach enables us, for the first time, to generate a robust training dataset along with a statistical algorithm for the retrieval, while bypassing the need for a land surface model and radiative transfer scheme. This new global product of surface turbulent fluxes is named WECANN (Water, Energy, and Carbon Cycle fluxes with Artificial Neural Networks). WECANN monthly flux estimates for the period 2007 – 2015

are provided on a $1° \times 1°$ resolution grid and with units of W m$^{-2}$ for LE and H, and gC m$^{-2}$ day$^{-1}$ for GPP.

A second key innovation of the WECANN methodology is that it uses the new remotely sensed SIF measurement as input. Previous studies show a strong relationship between the rate of photosynthesis and SIF observations and indicate that the plant fluorescence measurements can be a useful proxy for photosynthesis estimation (Flexas et al., 2002; Govindjee et al., 1981; Havaux and Lannoye, 1983; van Kooten and Snel, 1990; Krause and Weis, 1991; McFarlane et al., 1980; Toivonen and Vidaver, 1988; van der Tol et

al., 2009). Recently, satellite observations of SIF have become available, opening new possibilities for the global monitoring of photosynthesis (Frankenberg et al., 2011, 2012, 2014; Guanter et al., 2012; Joiner et al., 2013; Schimel et al., 2015; Xu et al., 2015). SIF observations from the Global Ozone Monitoring Experiment–2 (GOME-2) instrument are shown to be more sensitive to plant photosynthesis (both seasonal variability and intensity) compared to typical optical-based vegetation index estimates (such as the Enhanced Vegetation Index - EVI) (Joiner et al., 2011). Another SIF product retrieved from the Greenhouse gases Observing

SATellite (GOSAT) has been used to study the impact of seasonal variability on vegetation productivity in Amazon rainforest and shows that SIF is a pertinent indicator of vegetation water stress (Lee et al., 2013). Moreover, a strong linear relationship between GOSAT-based SIF retrievals and GPP is found for different vegetation types which suggests that SIF estimates can be combined with plant physiological fluorescence models for future global carbon cycle research (Frankenberg et al., 2011).

Recently, a new SIF product was developed from observations of the GOME-2 satellite using a new retrieval algorithm that

disentangles three components from  multispectral observations (Joiner et al., 2013). SIF retrievals are shown not to be strongly



affected by cloud contamination and seasonal variabilities in aerosol optical depth (Frankenberg et al., 2014). More recently, remotely sensed SIF retrievals have been used to successfully provide estimates of GPP in cropland and grassland ecosystems (Guanter et al., 2014; Zhang et al., 2016a). SIF retrievals are also integrated with photosynthesis estimates from National Center for Atmospheric Research Community Land Model version 4 (NCAR CLM4) which result in significant improvement of the

5 photosynthesis simulation (Lee et al., 2015). As GPP directly relates to plant transpiration through stomata regulation (Damour et al., 2010; DeLucia and Heckathorn, 1989; Dewar, 2002), and transpiration water fluxes dominate continental ET (Jasechko et al., 2013), the use of remotely sensed SIF has the potential to better constrain estimates of the continental water (LE), and energy (H) cycles, in addition to carbon (GPP) cycle.

The rest of the paper is organized as follows. The datasets used as input and target are introduced in Section 2. The ANN retrieval

and Bayesian characterization methods are explained in Section 3. Section 4 provides the results of flux retrievals, validation of results and discussions on the impact of SIF on the retrievals. Conclusions are presented in Section 5.

## 2 Data

This section provides details of each of the remote sensing and/or model-based estimates of the variables used as input or target data in the ANN framework, as well as the tower data used to validate the retrievals. The inputs to WECANN include six remotely

sensed variables introduced in Section 2.2: SIF, net radiation, air temperature, soil moisture, precipitation, and snow water equivalent. These are used to retrieve the three surface fluxes (LE, H, and GPP). Different observation and/or model based datasets are used as the training dataset, and are explained in Section 2.1. All the data presented here are projected and gridded on a $1° \times 1°$ geographic grid and averaged at monthly temporal resolution. The ocean and ice covered pixels were masked using the land mask data from National Snow and Ice Data Center (NSIDC) (Brodzik and Armstrong, 2013). Finally, the FLUXNET tower data used

for validation of the ANN retrievals are presented in Section 2.3.

### 2.1 Training Datasets

Four products are introduced in this section, and a triplet of them is used for training of each of the LE, H, and GPP (Section 3.2). For LE and H, training is performed based on GLEAM, FLUXNET-MTE, and ECWMF ERA HTESSEL. For GPP, training is performed on FLUXNET-MTE, ECWMF ERA HTESSEL, and MODIS-GPP. Table 1 summarizes the characteristics of the

25 training datasets used here.

#### 2.1.1 GLEAM

The Global Land Evaporation Amsterdam Model (GLEAM) is a set of algorithms to estimate terrestrial evapotranspiration using satellite observations (Martens et al., 2016; Miralles et al., 2011a). GLEAM is a physically-based model composed of 1) a rainfall interception scheme, driven by rainfall and vegetation cover observations; 2) a potential evaporation scheme, calculated from the

30 Priestley and Taylor (1972) equation and driven by satellite observations; and 3) a stress factor attenuating potential evaporation, based on a semi-empirical relationship between microwave VOD observations and root-zone soil moisture estimates (based on a running water balance for rainfall and assimilating satellite soil moisture). The data is provided on a $0.25° \times 0.25°$ spatial resolution and daily temporal resolution and starts in 1980. GLEAM data have been used for studying land-atmosphere interactions, and the global water cycle (Guillod et al., 2014, 2015, Miralles et al., 2011a, 2014a, 2014b). In this study, we use LE and H estimates from

35 the latest version v3.0a (Martens et al., 2016).



### 2.1.2 FLUXNET-MTE

The FLUXNET-MTE (Multi-Tree Ensemble) provides global surface fluxes at $0.5° \times 0.5°$ spatial resolution derived from empirical upscaling of eddy-covariance measurements from the FLUXNET global network (Baldocchi et al., 2001). The MTE method used is an ensemble learning algorithm that enables learning diverse sequence of different model trees by perturbing the base learning algorithm (Jung et al., 2009, 2010, 2011). The data covers the period from January 1982 to December 2012 and can be used for benchmarking land surface models and assessment of biosphere gas exchange. We use LE, H, and GPP estimates from FLUXNET-MTE.

### 2.1.3 ECMWF ERA HTESSEL

The European Centre for Medium-Range Weather Forecasts (ECMWF) Reanalysis (ERA) is a global 3D variational data assimilation (3DVAR) product that uses the Hydrology Tiled ECMWF Scheme for Surface Exchanges over Land (HTESSEL) in the forecast system. HTESSEL has a surface runoff component and accounts for a global non-uniform soil texture unlike the old TESSEL model (Balsamo et al., 2009). This is an offline model simulation, and HTESSEL is driven by meteorological forcing output from the forecast runs. Photosynthesis in the model is computed independently from LE, so that the carbon cycle does not interact with the water cycle at the stomata level, adding errors. We use LE, H, and GPP estimates from ERA HTESSEL provided on a $0.25° \times 0.25°$ geographic grid with daily temporal resolution.

### 2.1.4 MODIS-GPP

The Moderate Resolution Imaging Spectroradiometer (MODIS) sensor is onboard the sun-synchronous NASA satellites Terra (10:30 AM/PM overpasses) and Aqua (1:30 AM/PM overpasses). It provides 44 global data products (Justice et al., 2002) from 36 spectral bands including visible, infrared and thermal infrared spectrums to monitor and understand Earth surface: atmosphere, land and ocean processes. The MODIS GPP/NPP project (MOD17) provides gross/net primary production estimates covering the whole land surface and is useful for analyzing the global carbon cycle and monitoring environmental change. The MOD17 algorithm is based on a light-use efficiency approach proposed by (Monteith and Moss, 1977), which states that GPP is proportional to the product of incoming Photosynthetically Active Radiation (PAR), fraction of Absorbed PAR (fAPAR) and efficiency of radiation absorption in photosynthesis. We use the monthly MOD17A2 GPP product (Running et al., 2004; Zhao et al., 2005; Zhao and Running, 2010). MOD17A2 is available from 2000 until 2015, and provided on a $0.05° \times 0.05°$ spatial resolution.

### 2.2 Input Datasets

Six sets of observations are used as input to the WECANN retrieval algorithm. These are selected in a way to provide necessary physical constraints on the estimates from the ANN. Table 2 lists the characteristics of each of the datasets, and they are briefly introduced in the following.

### 2.2.1 Solar-Induced Fluorescence

The GOME-2 instrument is an optical spectrometer onboard Meteorological Operational Satellite Program, MetOp-A and MetOp-B satellites, which were launched by the European Space Agency (ESA). GOME-2 was designed to monitor atmospheric ozone profile as wells as other trace gases and water vapor content. It senses Earth backscatter radiance and solar irradiance at a 40×80 km spatial resolution. Recently, the retrieval of Solar-Induced chlorophyll Fluorescence (SIF) using GOME-2 observations in the 650-800 nm spectrum has been investigated (Frankenberg et al., 2011; Joiner et al., 2011). We use version 26 of the daily SIF



product that uses the MetOp-A GOME-2 channel 4 with a ~0.5 nm spectral resolution and wavelengths between 734 and 758 nm. SIF estimates are provided on a geographic grid with $0.5° \times 0.5°$ grid spacing.

### 2.2.2 Net Radiation

Net radiation is the main control of the rates of sensible and latent heat in wet environments and is closely related to PAR. The

Clouds and Earth's Radiation Energy System (CERES) is a suite of instruments which measure radiometric properties of solar reflected and Earth emitted radiation from the Top Of Atmosphere (TOA) to Earth surface, from three broadband channels at 0.3 – 100 $\mu m$. The CERES sensors are on board the Earth Observation Satellites (EOS) including Terra, Aqua and TRMM (Kato et al., 2013; Loeb et al., 2009). We use the net radiation estimates from Synoptic Radiative Fluxes and Clouds (SYN) product of CERES which are provided on a $1° \times 1°$ geographic grid with monthly time resolution.

### 2.2.3 Air Temperature

The Atmospheric Infrared Sounder (AIRS) is a high-resolution spectrometer onboard the NASA Aqua satellite launched in 2002. It provides hyperspectral (visible and thermal infrared) observations for monitoring process changes in the Earth's atmosphere and land surface, as well as for improving weather prediction. The AIRS instrument was designed to obtain atmospheric temperature and humidity profiles of every 1 km layer of the atmosphere. The accuracy of AIRS temperature observations is typically better

than 1°C in the lower troposphere under clear sky condition (Aumann et al., 2003). We use daily temperature estimates from the lowest layer of AIRS level-3 standard product that is provided on a $0.5° \times 0.5°$ geographic grid.

### 2.2.4 Surface Soil Moisture

The European Space Agency (ESA) Climate Change Initiative (CCI) program soil moisture (ESA CCI SM) is a multi-decadal (1980–2015) global satellite-observed surface soil moisture product. It merges observations from passive sensors (e.g., Scanning

Multichannel Microwave Radiometer (SMMR), Special Sensor Microwave/Imager (SSM/I), AMSR-E) and active ones (e.g., the European Remote Sensing (ERS), Advanced Scatterometer (ASCAT)), based on a triple collocation error characterization (Dorigo, et al., in reivew; Liu, Parinussa, et al., 2011; Liu et al., 2012; Wagner et al., 2012). Here, we use daily data from the latest version, v2.3. ESA CCI SM is provided on a $0.25° \times 0.25°$ geographic grid.

### 2.2.5 Precipitation

The Global Precipitation Climatology Project (GPCP) provides global daily precipitation estimates at $1° \times 1°$ spatial resolution from Oct. 1996 to near present (Huffman et al., 2001). Global precipitation estimates from infrared and microwave instruments are combined with monthly gauge measurements to produce the daily estimates. In this study, v1.2 of the one-Degree Daily (1DD) product of GPCP is used and daily estimates are aggregated to monthly scales. Several studies have evaluated the GPCP 1DD product at global or regional scales, and results show that it has high accuracy and good agreement with independent in situ

measurements and other global precipitation estimates (Gebremichael et al., 2005; Joshi et al., 2012; McPhee et al., 2005; Rubel et al., 2002).

### 2.2.6 Snow Water Equivalent

The GlobSnow project is developed by ESA, and provides long-term snow-related variables: Snow Water Equivalent (SWE) and areal Snow Extent (SE). It combines microwave-based retrievals of snow information (including Nimbus-7 SMMR, DMSP

F8/F11/F13/F17 SSM/I(S) observations) and ground based station data through a data assimilation process and provides the SWE





and SE products at different temporal resolutions: daily, weekly and monthly (Pulliainen, 2006). Here, we use v2 of the daily L3A SWE product which is posted on a 25 km × 25 km EASE grid.

### 2.3 Validation Dataset: Eddy-Covariance Flux Observations

FLUXNET is a network of regional micrometeorological tower sites, which measure turbulent flux exchanges (water vapor, energy
fluxes and carbon dioxide) between ecosystems and atmosphere (Baldocchi et al., 2001). FLUXNET comprises over 750 sites covering five continents. Measurements from the FLUXNET towers provide valuable information for validating satellite based retrievals of surface fluxes. In this study, FLUXNET measurements from the FLUXNET2015 dataset for 21 stations are used to validate the WECANN retrievals. These sites are selected to span a large climatic and biome gradient (details are provided in section 4.4).

## 3. Methodology

### 3.1 Artificial Neural Network Setup

We developed an ANN retrieval algorithm to estimate the surface fluxes (LE, H, and GPP) based on our six sets of input observations: SIF, net radiation, air temperature, soil moisture, precipitation, and SWE (as described in Section 2.2). The ANN used here is a feedforward network consisting of three layers: (1) an input layer that directly connects to the input data, (2) one
hidden layer and (3) an output layer that produces the 3 output estimates. The number of neurons in the input and output layer is determined by the number of input and output variables, whereas for the hidden layer it has to be chosen according to the complexity of the problem (see below). The neuron output from each layer is fed to neurons in the subsequent layer through weighted connections. Each neuron output is the weighted sum of its inputs plus a bias, which is then subjected to a transfer function. In this study, we chose a tangent sigmoid transfer function for neurons in the hidden layer and a linear transfer function
in the output layer. A schematic of the ANN architecture is provided in Fig. 1.

The training step of the ANN aims at estimating the weights for each of the neuron connections, such that the mismatch between the ANN outputs and target estimates is minimized. For this, we used the mean squared error (MSE) as the cost function and a backpropagation algorithm to adjust the ANN weights. During training, the target data is divided into three subsets: training, validation and testing constituting 60%, 20% and 20% of the target data, respectively. In each iteration, the convergence of the
training and validation estimates towards the target data is checked. When overfitting of the network weights to the training data occurs, the validation estimates start diverging from the target data and the training is stopped (early stopping). The weights from the last iteration before the occurrence of the divergence represent the final solution. The test data are used to assess the ANN performance after the training phase.

As an additional measure to avoid overfitting, we repeated the training for several ANN with an increasing number of neurons in
the hidden layer (1 to 15). For 1 to 5 neurons, the $R^2$ value between the target data and NN estimates increased with an increasing number of neurons. For more than 5 neurons, little change in the skill was observed when increasing the number of hidden layer neurons (Fig. S1). Thus an ANN with 5 hidden layer neurons represents the simplest ANN that can converge to a solution and model the non-linear relationship between the satellite inputs and the surface flux estimates.

To train the ANN, we used LE, H and GPP estimates from the years 2008-2010. The target dataset was generated through a triple
collocation based merging of triplets of the flux estimates introduced in Section 2.1 (details are discussed in Section 3.2). After completion of the training, the performance of the ANN and its ability to generalize was evaluated using the LE, H and GPP target





data from 2011. Finally, WECANN retrievals are validated against other global products and eddy covariance tower data. Results of these comparisons are presented in section 4.

### 3.2 Target Dataset: A Bayesian prior using Triple Collocation

One of the key issues in the design of an ANN to retrieve any geophysical variable is defining a good training dataset. One practice has been to use outputs from a land surface model as the target (Aires et al., 2005; Jiménez et al., 2013; Kolassa et al., 2013; Rodríguez-Fernández et al., 2015). However, all observations and models contain random errors and biases. Therefore, the retrieval based on the ANN exhibits most of the biases of the original training dataset even if the ANN is able to make corrections to its original training target data (e.g. correction of an imperfect seasonal cycle, as demonstrated by Jiménez et al., 2009). To address this issue, we use multiple datasets, which are sufficiently independent so that the training can learn from each dataset and benefit from all of them, synergistically. We implement a pseudo Bayesian training by weighting the occurrence of each training dataset by its likelihood.

To define this prior distribution, we use the triple collocation (TC) technique. TC is a method to estimate the Root Mean Square Errors (RMSE) (and, if desired, correlation coefficients) of three spatially and temporally collocated measurements by assuming a linear error model between the measurements (McColl et al., 2014; Stoffelen, 1998). This methodology has been widely used in error estimation of land and ocean parameters, such as wind speed, sea surface temperature, soil moisture, evaporation, precipitation, fAPAR, and in the rescaling of measurement systems to reference system for data assimilation purposes (Alemohammad et al., 2015; D'Odorico et al., 2014; Gruber et al., 2016; Hain et al., 2011; Lei et al., 2015; Miralles et al., 2010, 2011b; Parinussa et al., 2011), as well as in validating categorical variables such as the soil freeze/thaw state (McColl et al., 2016). The relationship between each measurement and the true value is assumed to follow a linear model:

$$X_i = \alpha_i + \beta_i t + \varepsilon_i \qquad i = 1,2,3 \tag{1}$$

where $X_i's$ are the measurements from the collocated system $i$ (e.g. remote sensing observation, model output, etc), $t$ is the true value, $\alpha_i$ and $\beta_i$ are the intercept and slope of the linear model, respectively. $\varepsilon_i$ is the random error in measurement $i$ and TC estimates the variance of this random variables in each measurement. By further assuming that the errors from the three measurements are uncorrelated $(Cov(\varepsilon_i, \varepsilon_j) = 0,$ for $i \neq j)$ and the errors are uncorrelated with the truth $(Cov(\varepsilon_i, t) = 0)$, the RMSE of each measurement error can be calculated as (McColl et al., 2014):

$$\begin{bmatrix} \sigma_{\varepsilon_1} \\ \sigma_{\varepsilon_2} \\ \sigma_{\varepsilon_3} \end{bmatrix} = \begin{bmatrix} \sqrt{Q_{11} - \frac{Q_{12}Q_{13}}{Q_{23}}} \\ \sqrt{Q_{22} - \frac{Q_{12}Q_{23}}{Q_{13}}} \\ \sqrt{Q_{33} - \frac{Q_{13}Q_{23}}{Q_{12}}} \end{bmatrix} \tag{2}$$

in which $Q_{ij}$ is the $(i^{th}, j^{th})$ element of the covariance matrix between the three measurements. Since the triplet of datasets used for training each of the fluxes (see Table 1) is derived through different semi-empirical approaches with different sources of errors, the assumption of uncorrelated errors is more likely to be met.

The TC errors from the surface fluxes are shown in Figs. S2-S4. The white regions represent missing retrievals or discarded negative estimates due to insufficient data record. For LE, high TC errors are found in the Amazon rainforest and tropical Africa



for GLEAM, in Amazon rainforest and the Sahel for ECMWF, in Indian peninsula for FLUXNET-MTE and in U.S. Great Plains for ECMWF and FLUXNET-MTE. For H, beside the aforementioned regions, high TC errors are also found in Southeast Asia for GLEAM and ECMWF, and in northern Canada for FLUXNET-MTE. For GPP, MODIS and ECMWF have the highest errors in Amazon rainforest, ECMWF and FLUXNET-MTE have relatively higher errors in US Great Plains, and all three products have

5 similar errors in Tropical Africa.

There are several likely causes for these errors. For the FLUXNET-MTE data, the regions which are not covered by (many) FLUXNET eddy-covariance stations may result in larger uncertainties, and those regions for which interception is a large component of the LE flux as well (Michel et al., 2016). For the GLEAM and ECMWF data thick vegetation generally induces biases compared to the satellite observations, especially in tropical regions (Anber et al., 2015).

Finally, we use the TC-based RMSE estimates at each pixel to compute the *a priori* probability ($P_i$) of selecting a particular dataset in each pixel, if that pixel is used as part of the training dataset:

$$P_i = \frac{\frac{1}{\sigma^2_{\varepsilon_i}}}{\sum_{i=1}^{3} \frac{1}{\sigma^2_{\varepsilon_i}}}  \qquad (3)$$

in which $P_i$ is the probability of selecting dataset $i$ when sampling from three measurements. We assume that these probabilities are time independent as we are limited by the currently available duration of the input data; however, future versions will explore

the use of seasonally varying probabilities.

## 4. Results and Discussion

### 4.1 Global Magnitude and Variability of Surface Fluxes

In this section, we present and compare the retrievals of LE, H and GPP fluxes for the year 2011, which was not included in the training step of WECANN, thus it is used here to evaluate the ANN fit to the target values.

Figure 2 illustrates the global average annual retrieved fluxes and scatterplots of flux retrievals vs target estimates. The spatial patterns of the WECANN retrievals are similar to expectations. The average global fluxes in 2011 are 36.26 W m$^{-2}$ for LE, 34.82 W m$^{-2}$ for H, and 2.20 gC m$^{-2}$ day $^{-1}$ for GPP. LE has the best R$^2$ (0.95) comparing to the other three flux variables H (R$^2$=0.89), and GPP (R$^2$=0.90). The Root Mean Squared Difference (RMSD) of each of the retrievals with respect to the target estimates is as following: for LE, RMSD = 11.13 W m$^{-2}$; for H, RMSD = 13.35 W m$^{-2}$; and for GPP, RMSD=1.23 gC m$^{-2}$ day $^{-1}$.

The seasonal variability and spatial pattern of the surface flux retrievals from 2011 (LE, H, GPP) are shown in Figs. 3 - 5. LE does not exhibit any variability over deserts, such as the Sahara and Arabian Peninsula, as expected (Fig. 3). Tropical regions exhibit subtle seasonal variability in LE, such as in the Amazon rainforest, Congo basin and Southeast Asia. These spatial variabilities in the seasonal cycle reflect changes in the radiation, temperature, water availability during the dry season, soil nutrient, soil type conditions as well as leaf flushing (Anber et al., 2015; Morton et al., 2014, 2016; Restrepo-Coupe et al., 2013; da Rocha et al.,

2009; Saleska et al., 2016). In contrast, seasonal variability dominated by radiation availability are noticeable in wet mid-latitude regions for both Northern and Southern Hemisphere, i.e., East Asia, Eastern U.S. and Australian North and East Coast with over 60 W m$^{-2}$ difference between winter and summer months. One exceptional case is South Asia, where LE does not significantly rise in spring, likely due to the effects of the monsoonal climate.

Seasonal variabilities in H (Fig. 4) are distributed in opposite pattern to LE, as expected. Deserts and dry regions i.e., the Sahara,

Southwestern U.S. and Western Australia demonstrate much more seasonal variability than the rest of the world, given the strong water limitations, the available energy converted into H becomes dictated by the seasonal cycle of solar radiation. In contrast, tropical rainforests (Amazon, Congo, Indonesia) exhibit limited seasonal variability. In mid-latitude energy-limited regions



(Central/Eastern Europe, Easter US), H also reflects the course of available energy, and in more water-limited regimes (e.g. Western US and Mediterranean Europe), it reflects the interplay between soil dryness and available energy, with a peak between spring and summer for dry regions.

The seasonal variability of GPP (Fig. 5) in Northern latitudes follow the availability of radiation in wet regions with a peak in summer and another in spring for dry regions, corresponding to both soil water availability and high incoming radiation. A clear East-West transition conditioned by water availability is observed in continental U.S. In tropics and subtropics, the response is diverse. The Amazon rainforest exhibits high GPP throughout the year with a peak between September and February in the wetter part of the basin, following the dry season, consistent with the observations at eddy-covariance towers near Manaus and Santarem (Restrepo-Coupe et al., 2013; da Rocha et al., 2009). Compared to LE, substantial geographical variability are observed in the Amazon, because of the strong variabilities in soil type, green up, biodiversity and soil water availability. In the drier part of the basin, water availability controls the seasonal cycle of photosynthesis and the peak in GPP is observed in the wet season (DJFMA). In the Congo rainforest, GPP exhibits four seasons, with two wet and two dry ones, with substantial decrease in GPP during those dry spells. In Indonesia, GPP is steadier throughout the year, exhibiting high values year round. Monsoonal climates over India, South-East Asia, Northern Australia and Central-Northern America are well captured with rapid rise in GPP following water availability. The highest GPP are observed in rainforests and the US agricultural Great Plains, in JJA for the latter. Northern latitude regions mainly exhibit substantial GPP in the summer and late spring, and small values throughout the rest of the year.

### 4.2 Impact of SIF on the retrieval of surface fluxes

Satellite SIF observations are relatively new, and have not been used to estimate LE and H at global scales before. Therefore, we assess the information content of SIF observations in the WECANN retrievals by excluding them from the ANN inputs. We trained an ANN without SIF data on each of the three fluxes and evaluated the difference between the retrieved fluxes. Figure 6 shows the percentage difference maps between flux retrievals trained with SIF and without SIF for the year 2011 as well as the scatter plots with respect to target dataset. Including SIF decreases LE estimates in parts of Australia, Central Asia, Tibetan plateau and Southern Africa. In Indian peninsula, the Sahel, Eastern US and Southeastern Asia LE tends to reduce by adding the SIF information. Comparison of the ANN retrieval without SIF compared to the target data shows that the overall statistics of the ANN retrieval are comparable to WECANN retrievals, and inclusion of SIF slightly improves $R^2$ and lowers RMSD (Fig. 6d, and Table 3). Including SIF decreases H in the Sahel, Arabian Peninsula, Europe, Eastern US and in most of South America. In most of the other regions H is increased when SIF is added. The global ANN fits against the target H are relatively similar with and without SIF (Fig. 6e, Table 3).

Including SIF increases GPP in Central US, as well as in Europe, Northern India, and Southern Brazil capturing intense cropping regions. In Northern Canada, Central Asia, Australia, Southern Africa and the Tibetan Plateau GPP is strongly reduced by adding SIF into the ANN retrieval. In the Congo and Amazon, photosynthesis is slightly increased locally by the inclusion of SIF. Similarly to the other retrievals the global statistics of the retrieval with and without SIF compared to the target are relatively similar (Fig. 6f, and Table 3), hiding some of the changes in the spatial structure.

This comparison shows the significant role that SIF estimates play in the flux retrievals from WECANN. Given that GOME-2 instrument was originally designed to measure ozone in the atmosphere and not SIF, the future estimates of SIF from designated missions such as Fluorescence Explorer (FLEX) will have higher accuracy and finer spatial and temporal resolution (Kraft et al., 2012). Those SIF estimates will further enhance the retrievals of surface fluxes.



### 4.3 Comparison against other remote-sensing based products

In this section, we compare the WECANN-based estimates to other datasets. Figure 7 shows the comparisons for LE, and indicates that our product has higher $R^2$ with FLUXNET-MTE ($R^2 = 0.96$) and ECMWF ($R^2 = 0.96$) than with GLEAM ($R^2 = 0.94$). However, the scatterplot with FLUXNET-MTE is more concentrated and aligned along the 1:1 line, further emphasizing the consistency between the two datasets. Difference in spatial patterns shown in Fig. 7a-c reflect that WECANN exhibits smaller spatial differences with FLUXNET-MTE than GLEAM or ECMWF and such differences exhibit a narrower range between -10 and 10 W m$^{-2}$. FLUXNET-MTE overestimates LE compared to our product in transitional tropical and subtropical regions and particularly over India, which are regions with few eddy-covariance towers. GLEAM exhibits substantial differences with our product particularly in regions dominated by seasonal water stress such as Brazilian savannas, the Horn of Africa, Central America, India and the subtropical humid part of Africa south of the Congo. In the Sahel, GLEAM LE is higher than our estimate and FLUXNET-MTE. The LE estimate of ECMWF is nearly always higher than our estimate with much higher values in the Congo, the Amazon, Southern Brazil, and Northern Canada. In Europe, where the ECMWF estimate should be best because of the frequent weather operational forecast checks and model adjustment in the region, the estimates are more similar. The differences and similarities of WECANN retrievals with the three target datasets is consistent with the error estimates from TC. For example, Fig. S2 shows that FLUXNET-MTE has the smallest error in LE estimates globally compared to GLEAM and ECMWF, other than across India. WECANN retrievals also have better agreement with FLUXNEWT-MTE.

The differences in H estimates are more complex (Fig. 8). First, the $R^2$ between WECANN and the other datasets is always lower than for LE. ECMWF and FLUXNET-MTE again yield higher $R^2$ with WECANN (0.85 and 0.84, respectively) while GLEAM has an $R^2$ of 0.80. GLEAM exhibits lower H in most of the Northern hemisphere, especially in seasonally dry regions, potentially due to its simple formulation of G. H estimates are relatively higher over the Amazon and Congo but lower over Indonesia for GLEAM. In the Southern Sahara and northern Sahel as well as in Eastern Asia and Canada GLEAM has lower H compared to WECANN and FLUXNET MTE. ECMWF exhibits higher values in seasonal dry regions such as Western US, Brazilian Savannas, Southern Congo, the Sahel compared to WECANN and smaller values in the Amazon, Indonesia, and over desert areas of the Sahara and Arabic peninsula as well as South East Asia. The GLEAM and ECMWF H difference maps show many similar patterns: the Sahara, Eastern Europe, East Asia are underestimated, while Southern Africa and Eastern part of Amazon are overestimated. Similarly the errors patterns estimated from TC (Fig. S3) are consistent with the comparison of WECANN and target datasets. Figure S3 shows that ECMWF has higher errors in the Sahel, Southern Congo, and Brazilian Savana and GLEAM has higher errors in the Amazon, East Asia and Central Africa.

The comparison between the GPP estimates shows significant differences (Fig. 9). WECANN compares the best against FLUXNET-MTE ($R^2 = 0.92$), with MODIS ($R^2 = 0.90$) and ECMWF ($R^2 = 0.87$) following. While FLUXNET-MTE and MODIS have similar $R^2$, their spatial differences are distinct. In the Amazon, ECMWF and FLUXNET-MTE have larger GPP estimates compared to WECANN, while MODIS estimates are much smaller. In cold northern latitude regions of Siberia and Northern Canada, all three products have higher GPP than WECANN. In Congo, MODIS and FLUXNET-MTE have higher GPP, while ECMWF has a lower one. In Central and Southwestern US, all three products tend to yield lower GPP. Comparison of these findings with the error estimates from TC (Fig. S4) shows that FLUXNET-MTE has the lowest errors globally, while ECMWF has the largest errors in the Amazon.

### 4.4 Validation with FLUXNET Data

Direct validation of the WECANN fluxes is made more challenging by the fact that no global, error-free flux estimates are available. Remote sensing or model products such as those used for training have their own errors. In situ estimates from eddy covariance



towers with a footprint of a few 100 m may not be representative of the entire $1° \times 1°$ pixel, and are known to have problems with energy closure. When three datasets with uncorrelated errors (commonly assumed to be true if the sources of error in each dataset have no common physical origin) are available, triple collocation provides a valuable technique to validate large-scale datasets in the absence of a known truth. However, WECANN's use of different noisy training datasets may cause the presence of some

correlated errors between WECANN fluxes and other possible large-scale triple collocation inputs. Instead, we validate the fluxes by comparing them to data from several FLUXNET eddy-covariance towers. However, it is important to keep in mind that these flux estimates may themselves have errors relative to the true 1-degree scale fluxes and their footprint not be representative of the WECANN $1° \times 1°$ pixels.

We compare the model outputs to eddy-covariance towers from the FLUXNET 2015 database (tier 1 and tier 2,

http://fluxnet.fluxdata.org/data/fluxnet2015-dataset/) spanning a large climatic and biome gradient (Fig. 10). The data have been systematically quality controlled with a standard format throughout the dataset (http://fluxnet.fluxdata.org/data/fluxnet2015-dataset/data-processing/, (Pastorello et al., 2014)) and gap-filled using ERA meteorological forcing downscaling. The NEE is partitioned as the sum of Gross Primary Production (GPP) and Ecosystem Respiration (RECO) using one of two methods. The first method is based on the extrapolation of nighttime data (Reichstein et al., 2005), which is used to parameterize a respiration

model that is then applied over the daytime NEE to estimate RECO. GPP is then calculated as the difference between RECO and NEE. The second method uses daytime data to parameterize a model of both GPP and RECO (Lasslop et al., 2010). The partitioning method used varies from site to site.

In addition to the FLUXNET 2015 dataset, we use data from the Large-scale Biosphere-Atmosphere (LBA) experiment in Brazil. Specifically, we use data from sites near Santarem, Pará (Site code BR-Sa3), in Rondônia at the edge of a deforested region (BR-

Ji1 and BR-Ji2) and near São-Paulo (BR-Sp1). As the data did not span recent years we instead use a climatology of the fluxes for comparison. We note that, of course, the inter-annual variability in the region (such as El Niño and La Niña) could alter the seasonality and magnitude of the fluxes in the region.

A summary of statistics across the different sites combining the FLUXNET 2015 tier 1 database is provided in Table 4-Table 6. Overall, WECANN performs better than the alternative global products. In particular, WECANN has the highest correlation for

61% of sites for LE, 60% of sites for H, and 56% of sites for GPP. This high $R^2$ reflects the capacity of WECANN to correctly capture the seasonal cycle and interannual variability. One of the reasons for this is the presence of the SIF information in the ANN retrieval, which is directly related to GPP and plant transpiration, contrary to optical vegetation indices that are sensitive to vegetation greenness and canopy cover - factors which can lag fluxes or be out of phase (see e.g. the lower correlation with NDVI in Frankenberg *et al.*, 2011). The RMSE of WECANN is lower than all other products at 56% of sites for LE, 50% of sites for H,

and 44% of the sites for GPP. The bias is also reduced compared to other retrievals, even if some variability can be seen from site to site.

Figure 11 shows the comparison of monthly WECANN retrievals with the tower estimates across 5 European sites. At the AT-Neu site, Neusflit, Stubai Valley, Austria (Fig. 11a), the seasonal cycle is correctly captured for both LE and GPP. All flux retrievals perform relatively well at this site dominated by radiation and temperature. The GPP based on the eddy covariance has a sharper

and earlier rise in the spring than LE, which seems unrealistic and may be an artifact of the GPP retrieval method. WECANN is slightly delayed compared to the observed LE, possibly a reflection of the larger footprint encapsulating various conditions in this steep topography region. All flux retrievals overestimate the H observations, even though they capture some of the seasonality. The observed H lags the observed LE, which seems unrealistic given that the region is mostly radiation limited so that a spring increase in radiation and temperature should affect both fluxes. The large footprint of the retrieval could be another source of error,

as it would sample multiple environmental conditions. Nonetheless, the ECMWF and GLEAM retrievals are the closest to the



observed H and FLUXNET-MTE strongly overestimates the observed flux, similarly to WECANN, even though the bias is not as high.

At the Brasschaat site, BE-Bra, Belgium (Fig. 11b), all retrievals strongly underestimate the reported eddy-covariance H. At this humid site though, the magnitude of the measured H is often higher or on the same order in the summer as LE. Given the high

degree of urbanization around the site, it is most likely a reflection of the footprint of the eddy-covariance and the fact that it observes urbanized surfaces with high H. Indeed the surface energy budget is not locally balanced and turbulent fluxes are higher than the observed net radiation minus ground heat flux. LE is very well captured by WECANN, which captures the seasonal cycle well, yet misses some of the interannual variability. WECANN outperforms the other retrievals of LE and GPP. WECANN captures the GPP seasonal cycle compared to other products, which display too early GPP rise and overestimate the summer GPP. Again,

the SIF data provides independent useful data compared to other environmental information (radiation, temperature, vegetation indices) used by the other retrieval schemes.

At another seasonally cold site, in Switzerland, CH-Fru (Fig. 11c), WECANN again performs very well, correctly reproducing the seasonality of all fluxes, especially compared to the other products, which tend to rise too early in the spring. The magnitude of H and LE is very similar to the observations, yet GPP seems to be overestimated by WECANN, yet much less so than other products.

At the Mediterranean, Spanish site, ES-LgS (Fig. 11d) WECANN correctly reproduces H and LE yet overestimates the magnitude of GPP, even though it correctly captures its seasonal dynamics. We note; however, that the region is highly heterogeneous both in terms of topography and vegetation coverage and that the site is located at some of the driest location of the region.

At the cold Finland site (FI-Hyy), WECANN very well captures the seasonal cycle of GPP and LE, as well as to a less extent of H. WECANN better reproduces the seasonality, amplitude and interannual variability compared to other retrievals (Fig. 11e).

At the Brazilian sites, spanning the Savanna region to the Amazonian rainforest (Fig. 12), we only consider the climatology of the results, as most the data (ending in 2006) was not available during the GOME-2 satellite period. We acknowledge potential differences when considering the climatology of the fluxes, as interannual variability could modify the derived climatological seasonality. At the Rondônia sites Ji1, all flux retrievals tend to overestimate LE and GPP. This is most likely a reflection of the large landscape fragmentation with deforested and non-deforested patches. Similarly, the dryness perceived at the flux tower is not

seen by most of the retrievals as forests can sustain photosynthesis during the dry season through deeper roots (da Rocha et al., 2009). At the nearby Ji2 site, on the other hand, most flux retrievals perform much better and correspondingly report a maintained GPP and LE in the dry season. GLEAM as well as ECMWF exaggerate the seasonal cycle of LE and H. WECANN is positively biased in H but correctly reproduces LE. FLUXNET-MTE better reproduces GPP than WECANN and both products outperform MODIS and the ECMWF retrievals. Relatively similar results are obtained at the wet Santarem site, Sa3, where both WECANN

and FLUXNET-MTE perform well in reproducing all fluxes. ECMWF and MODIS show the incorrect seasonality of the fluxes at the site, as GPP at the site reflects subtle leaf aging and flushing (Lopes et al., 2016; Saleska et al., 2016; Wehr et al., 2016), and radiation structure not captured by those models (Anber et al., 2015; Morton et al., 2014, 2016). At the other site near Sao Paulo, with dry winter savanna, most flux retrievals correctly capture the seasonal cycle, yet most retrievals and especially WECANN are in seasonal advance over the observed eddy covariance with a too early increase in GPP and LE. The site is located in a highly

heterogeneous agricultural landscape yet observes an evergreen broadleaf forest, which is not representative of the heterogeneous landscape seen by the remote sensing products.

In Canada, (Fig. 13), WECANN very well reproduces the seasonal cycle of LE, especially compared to the other products that produce a too early rise in LE during the spring season. WECANN also better reproduces the seasonal cycle of GPP compared to other products. Nonetheless, all GPP retrievals underestimate the reported eddy covariance GPP. This is true of both sites Qfo and

Qcu. The reported eddy-covariance GPP appears very small though, especially given the LE magnitude in the summer, pointing



to potential problem in the magnitude of the surface fluxes, which is drastically impacted by the high-frequency corrections of the turbulent co-spectrum and its parameterization (Mamadou et al., 2016). H is well reproduced by WECANN at the Qcu site, but the Qfo site exhibits nearly twice the H magnitude of the Qcu site in the summer. This does not appear realistic given that the radiative and LE conditions are relatively similar at the two sites. WECANN again better reproduces the seasonal cycle compared to the other products.

Across the continental US Ameriflux sites (Fig. 14), WECANN performs well in terms of seasonal and interannual dynamics. At the Oklahoma agricultural site (US-ARM), H and LE are well reproduced, yet dry year H is underestimated (Fig. 14a). The GPP reported at the site very rapidly decays at the end of the spring whereas the region is highly agricultural with sustained agriculture in the summer. The difference between the reported GPP and WECANN retrievals might be again due to the difference in the footprint of the two estimates, At the Illinois site, US-Ib2, the dynamics of LE is relatively well reproduced by most products except for ECMWF (Fig. 14b). All retrievals overestimate GPP, especially FLUXNET-MTE. WECANN exhibits a late delay in the GPP decay. The measured H is very noisy yet exhibits a summer decay which is only partially captured by the different products. At the evergreen needleleaf Maine site, US-Me2, WECANN reproduces the dynamics of H, LE and GPP well, even if it underestimates the peak fluxes (Fig. 14c). Over the irrigated maize site in Nebraska (US-Ne1), the retrievals underestimate the peak LE and GPP, as well as overestimate the H in the peak summer season (Fig. 14d). This is most likely a reflection of the larger area observed or modeled by the flux retrievals which do not include similar intensive irrigation practices, leading to lower peak LE (and correspondingly higher H) and GPP. Only FLUXNET-MTE reproduces the magnitude of this irrigated site (but US-Ne1 was included in the FLUXNET-MTE training database). Finally, at the monsoonal grassland site of Santa Rita, AZ, WECANN correctly captures the complex dynamics of H and LE at the site with sometimes rain periods preceding the Monsoon period (Fig. 14e). Yet, WECANN slightly underestimates LE and overestimates GPP. In fact, most flux retrieval overestimate GPP in the dry and cold seasons. The landscape in the region is highly heterogeneous with denser vegetation in riparian zones, away from the tower location, which may explain the lower GPP value at the site compared to estimates of the larger-scale values.

Figure 15 shows the comparison of retrievals at two other sites. At the Daly River pasture, AU-DaP, Australia (Fig. 15a), WECANN reproduces very well the observed LE in terms of both seasonal and interannual variability. Compared to other products, WECANN better reproduces the seasonal cycle of this Monsoonal site, with a rapid rise in LE and lagged drying. Most retrievals fail to correctly reproduce the exact H seasonality, which is in opposite phase with LE, at this water limited site. All retrievals tend to overestimate the retrieved eddy-covariance GPP and fail to correctly capture the rapid rise in GPP, except for WECANN. The eddy-covariance GPP decay occurs significantly in advance over the LE decay. It seems unlikely that during the drying phase soil evaporation would explain nearly all of the LE and that transpiration would be so small (as indicated by the drop in GPP before LE). It is most likely due to an artifact in the model fitting of the respiration component, which implicitly assumes some stationarity. Nonetheless, all remote sensing retrievals seem to overestimate the dry season GPP.

At the South African Mediterranean site, ZA-Kru, WECANN reproduces some of the dynamics of the observed H, yet is typically smoother (Fig. 15b). Similarly, it reasonably captures the LE dynamics, except for the suspect cold season increase reported at the tower in 2013 (like other products). All products overestimate the reported GPP, though WECANN is closest to the observations and better captures the seasonal dynamics.

Overall, across the different sites, the WECANN retrieval performs better than other products, especially in terms of the seasonality of the fluxes. Several factors contribute to the capability of WECANN in having a better retrieval compared to other products. The ANN approach in WECANN uses a novel training technique to remove highly uncertain and outlier estimates from its target dataset. Therefore, WECANN retrievals are closer to the truth than each of the single target datasets. Moreover, the SIF measurements that are directly correlated with GPP provide a better constraint on flux estimates.



**5 Conclusion**

This study introduces a new statistical approach to retrieve global surface latent and sensible heat fluxes as well as gross primary productivity using remotely sensed observations at a monthly time scale. The methodology is developed based on an Artificial Neural Network (ANN) that uses six input datasets including solar induced fluorescence (SIF), precipitation, net radiation, soil

moisture, snow water equivalent, and air temperature. Moreover a Bayesian approach is implemented to optimally integrate information from three target datasets for training the ANN using Triple Collocation to calculate *a priori* probabilities for each of the three target datasets based on their uncertainty estimates.

The new global product, referred to as WECANN, is validated using target datasets as well as FLUXNET tower observations. The validation results comparing with target outputs show that our retrieval is best correlated with FLUXNET-MTE for LE ($R^2$=0.96),

H ($R^2$=0.84) and GPP ($R^2$=0.92), which is believed to be one of the most realistic global datasets and it has the lowest RMSE based on our TC error estimates (Fig. S2 – Fig. S4), despite its reported underestimated inter-annual variability due to the use of climatological values for several meteorological drivers (Miralles et al., 2014a, 2016). Such tendency also can be summarized from the global difference maps, which show that FLUXNET-MTE has the best agreement with WECANN retrievals. The WECANN and FLUXNET-MTE approaches are both based on machine learning, although the FLUXNET-MTE retrievals use a regression

tree rather than an ANN. Nevertheless, this commonality of methods may also contribute to the greater correspondence between these two datasets.

The flux retrieval maps indicate that all three fluxes have similar seasonal variability and distribution which are determined by annual phenological cycle in energy limited Northern latitude regions, dryness in Mediterranean and Monsoonal climates and by light availability in rainforests. Seasonal radiation has great impact on some regions for all flux variables, such as Eastern U.S.,

Europe and East Asia, which have wet conditions, are highly vegetated and located in mid-latitudes. As opposed to this, the seasonal variability for all fluxes in some low-latitude and wet condition regions, such as Amazon rainforest, Southern Africa and Southeast Asia, as well as some low-latitude arid regions, such as Southwest U.S., Western Australia, North Africa and Western Asia are not significant, as there is less seasonal solar radiation variability in aforementioned regions. Comparison between the flux variables LE, H, and GPP, they all demonstrate generally similar patterns of seasonal variability through time.

We also assessed the impact of SIF on retrieval quality. The difference maps between neural network outputs trained with SIF and without SIF demonstrate that SIF has high influence on all three flux retrievals (Fig. 6).

Finally, from the validation results comparing with FLUXNET tower observations, it is noted that WECANN has better performance compared to other global products. LE and H estimates from WECANN are more consistent with tower observations compared to GPP. WECANN retrievals have better correlation with tower observations in 61% of site for LE, 60% of sites for H,

and 56% of sites for GPP compared to other products. Moreover, retrievals from WECANN outperform other global products in capturing the seasonality of surface fluxes across a wide range of sites with different climatic and biome conditions.

**Data Availability**

WECANN product is available for free upon request. Please contact the corresponding author to request access.

**Competing Interests**

The authors declare that they have no conflict of interest.



**Acknowledgments**

The funding for this study is provided by the NASA grant # NNX15AB30G. PG acknowledges funding from NSF CAREER Award # EAR - 1552304, and NASA grant # 14-AIST14-0096. DM and PG acknowledge funding from the Belgian Science Policy Office (BELSPO) in the frame of the STEREO III programme project STR3S (SR/02/329). The authors would like to thank all the
5 producers and distributors of the data used in this study. The ECMWF team (Dr. Gianpaolo Balsamo and Dr. Souhail Bousetta, in particular) for providing the ECMWF data. We also thank NASA and Prof. Running for providing the MODIS GPP estimates and Dr. Johanna Joiner for the GOME-2 data. The GPCP 1DD data were provided by the NASA/Goddard Space Flight Center's Mesoscale Atmospheric Processes Laboratory, which develops and computes the 1DD as a contribution to the GEWEX Global Precipitation Climatology Project. This work used eddy covariance data acquired and shared by the FLUXNET community,
including these networks: AmeriFlux, AfriFlux, AsiaFlux, CarboAfrica, CarboEuropeIP, CarboItaly, CarboMont, ChinaFlux, Fluxnet-Canada, GreenGrass, ICOS, KoFlux, LBA, NECC, OzFlux-TERN, TCOS-Siberia, and USCCC. The FLUXNET eddy covariance data processing and harmonization was carried out by the ICOS Ecosystem Thematic Center, AmeriFlux Management Project and Fluxdata project of FLUXNET, with the support of CDIAC, and the OzFlux, ChinaFlux and AsiaFlux offices.

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




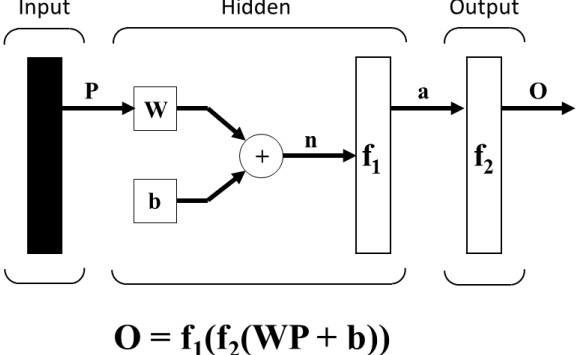

$$O = f_1(f_2(WP + b))$$

**Figure 1: Architecture of the ANN layers. Input layer provides the matrix P of the inputs to the Hidden layer. Hidden layer has a matrix W of weights and b of biases for the neurons, and the f₁ transfer function. The output of the Hidden layer (a = f₁(WP +b) ) is an input to the Output layer that applies the transfer function f₂ to the estimates and generates final outputs O.**

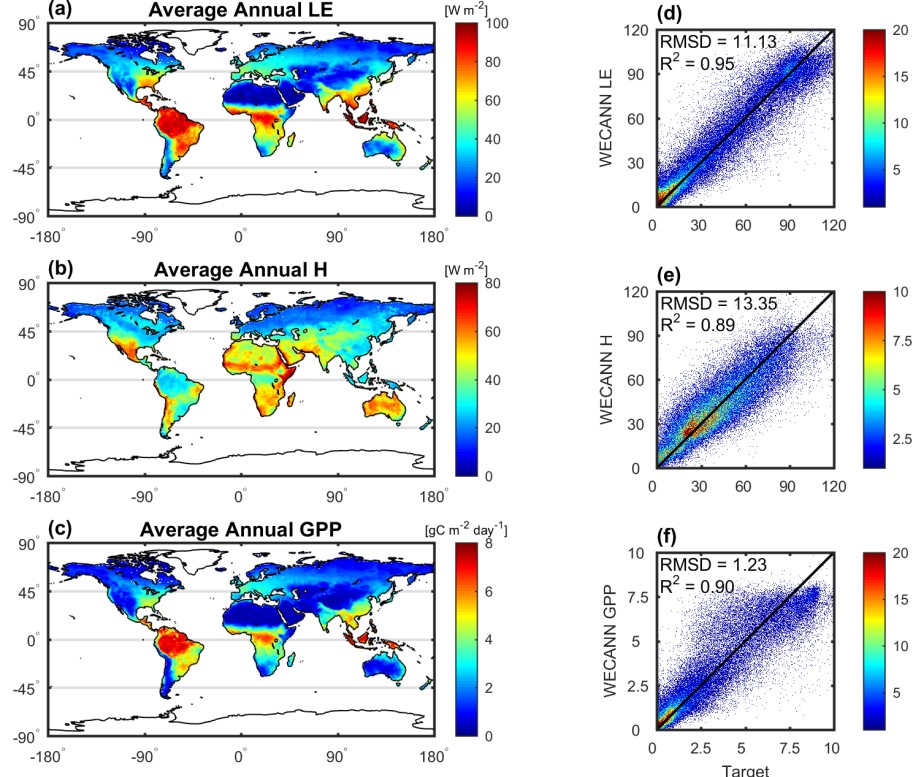

**Figure 2: Left column: Annual average surface fluxes in 2011 for (a) LE, (b) H, and (c) GPP. Right column: Density scatterplot between estimates of ANN and target data for (d) LE, (e) H, and (f) GPP during the validation period (2011). The density of scatter points is represented by the shading color. The diagonal black line depicts the 1:1 relationship.**





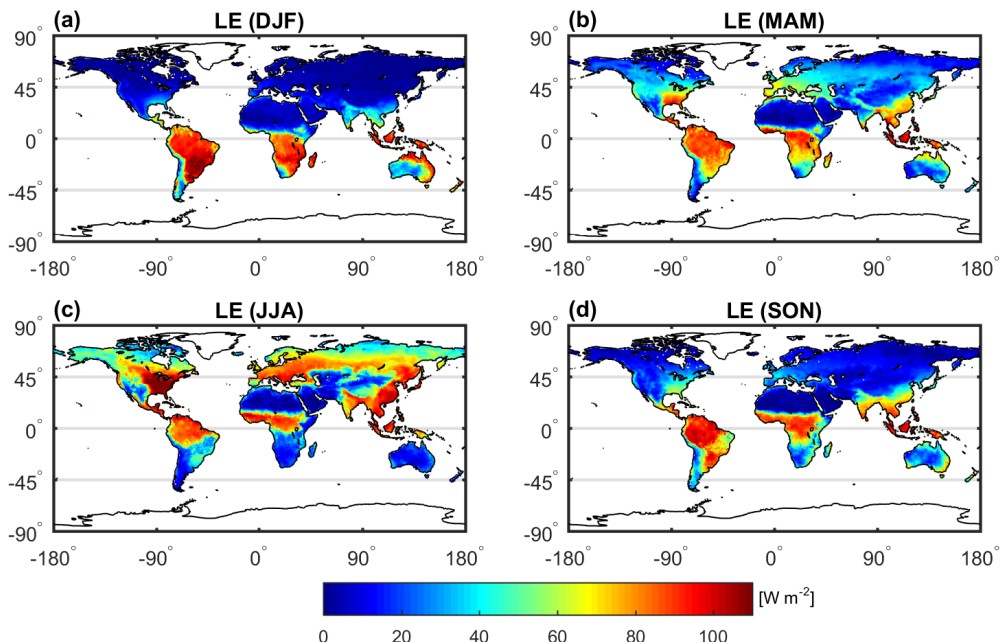

**Figure 3: Global patterns of seasonal average LE from WECANN in 2011, (a) December - February, (b) March - May, (c) June - August, and (d) September - November.**

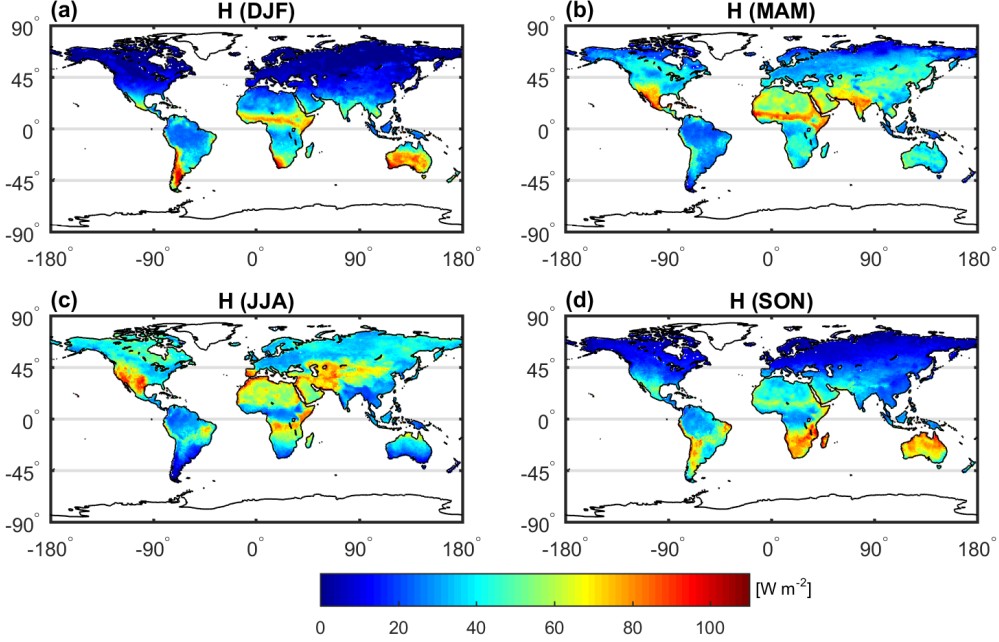

5    **Figure 4: Similar to Figure 3 but for H instead of LE**





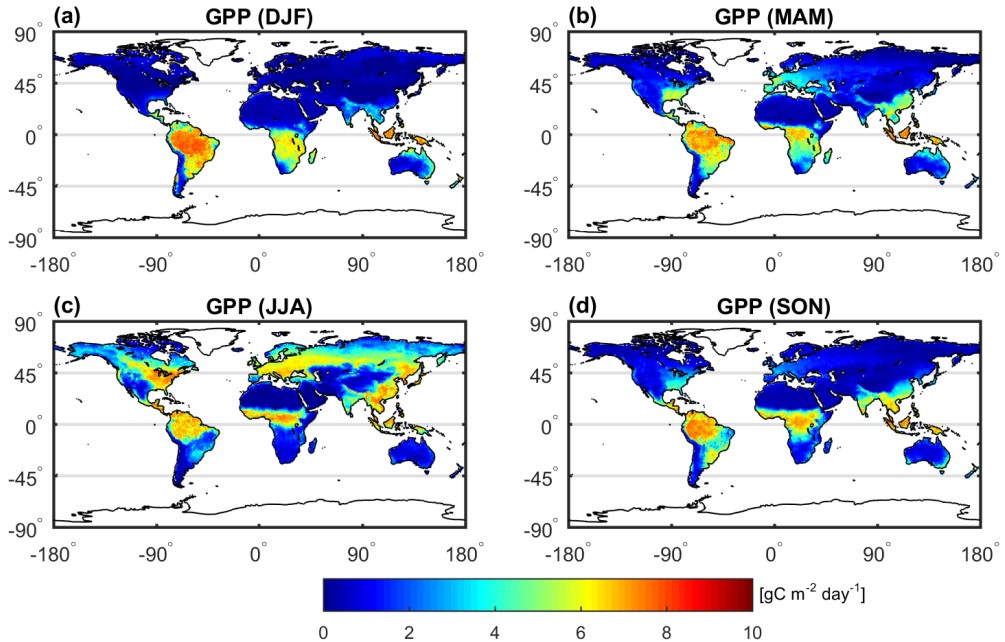

**Figure 5: Similar to Figure 3 but for GPP instead of LE**

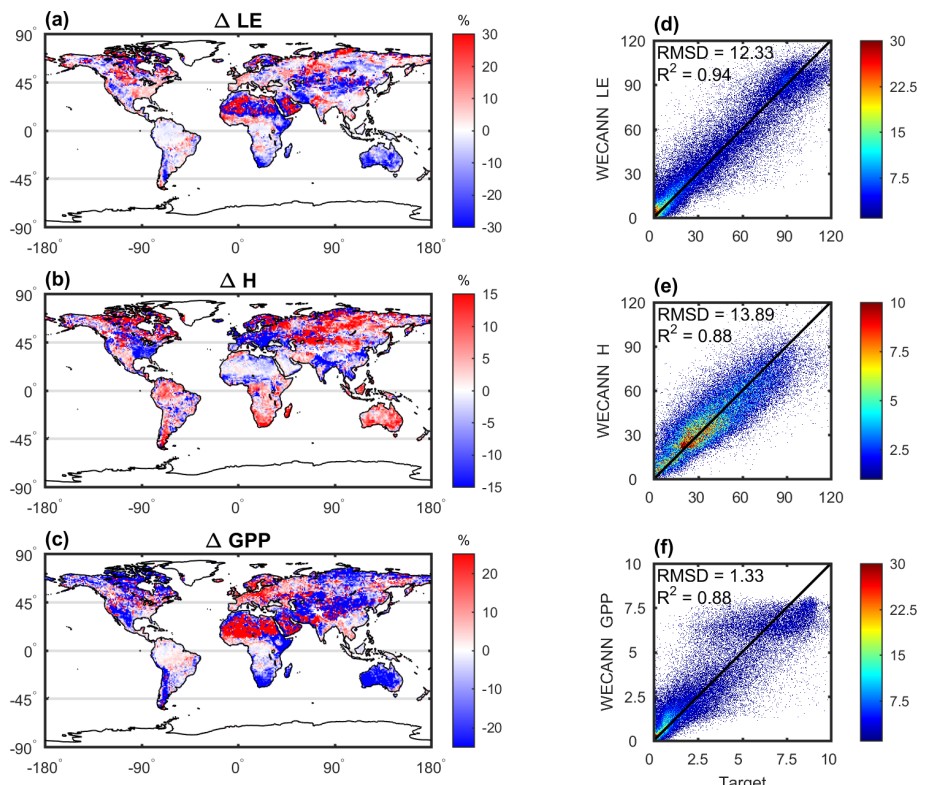

**Figure 6: Comparison of WECANN retrievals with the retrievals from an ANN without using SIF data. (a) – (c) shows the WECANN retrieval minus the retrieval without SIF normalized by the WECANN retrievals for LE, H, and GPP during 2011, respectively. (d)- (f) show the scatter plots of WECANN retrievals vs target data.**





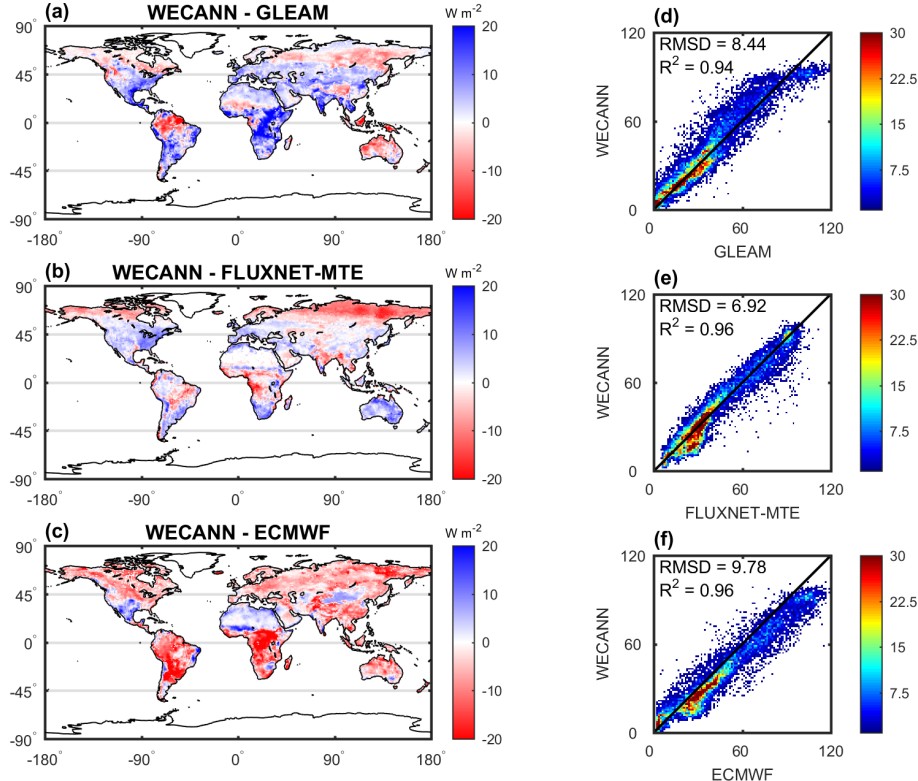

**Figure 7: Difference between annual mean LE retrieved by WECANN and the three target datasets (a-c). Scatter plots of LE retrieved**
5   **from WECANN vs. from each of the target datasets (d-f). Data used are from 2011.**





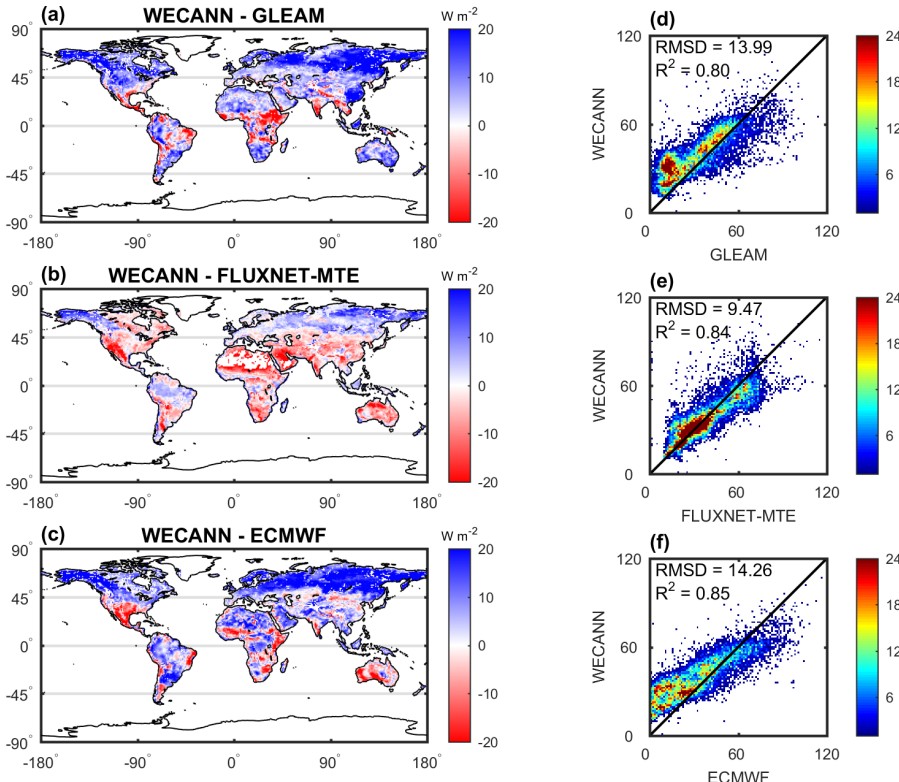

Figure 8: Similar to Figure 7 but for H instead of LE





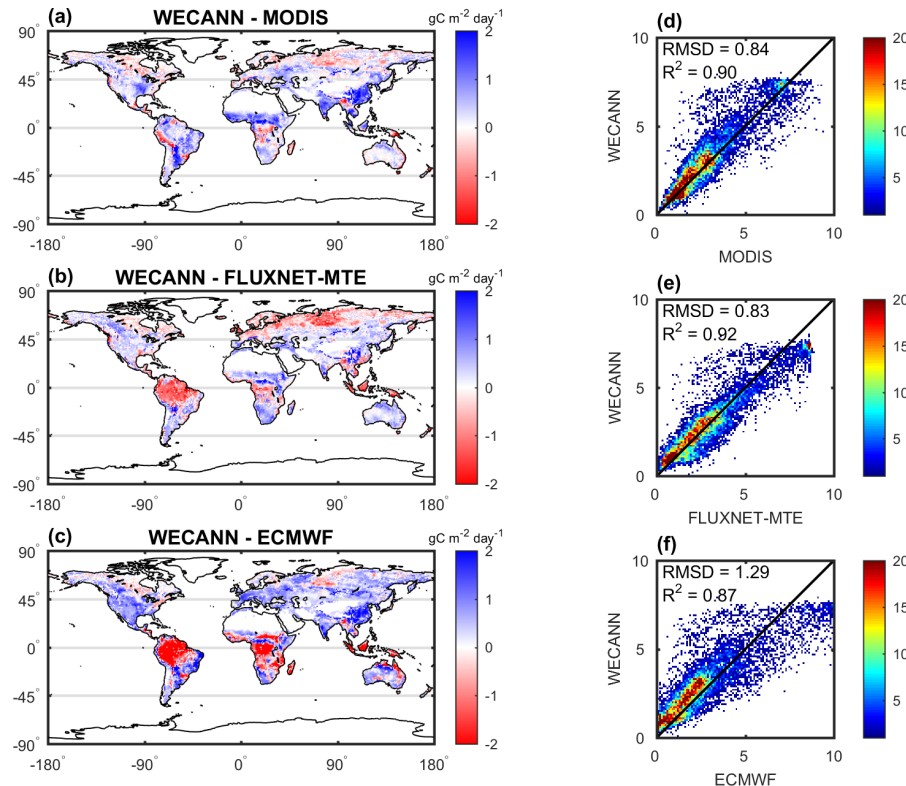

Figure 9: Similar to Figure 7 but for GPP instead of LE

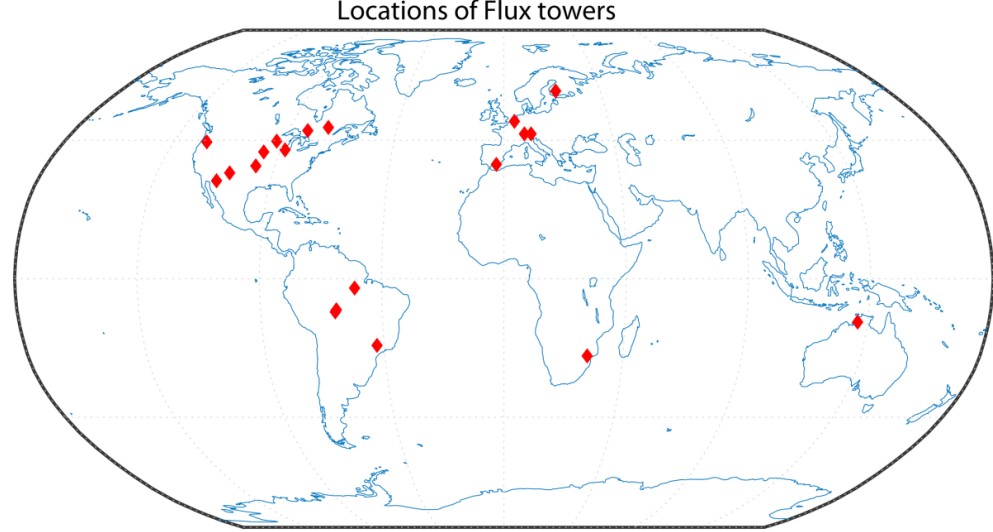

5    Figure 10: Geographical positions of the eddy-covariance sites used for comparison of the flux retrievals.







**Figure 11: Comparison of the flux retrievals with eddy covariance observations of LE, H and GPP across European sites (a) AT-Neu site, Austria, (b) BE-Bra site, Belgium, (c) CH-Fru site, Switzerland, (d) ES-LgS site, Spain, and (e) FI-Hyy site, Finland**





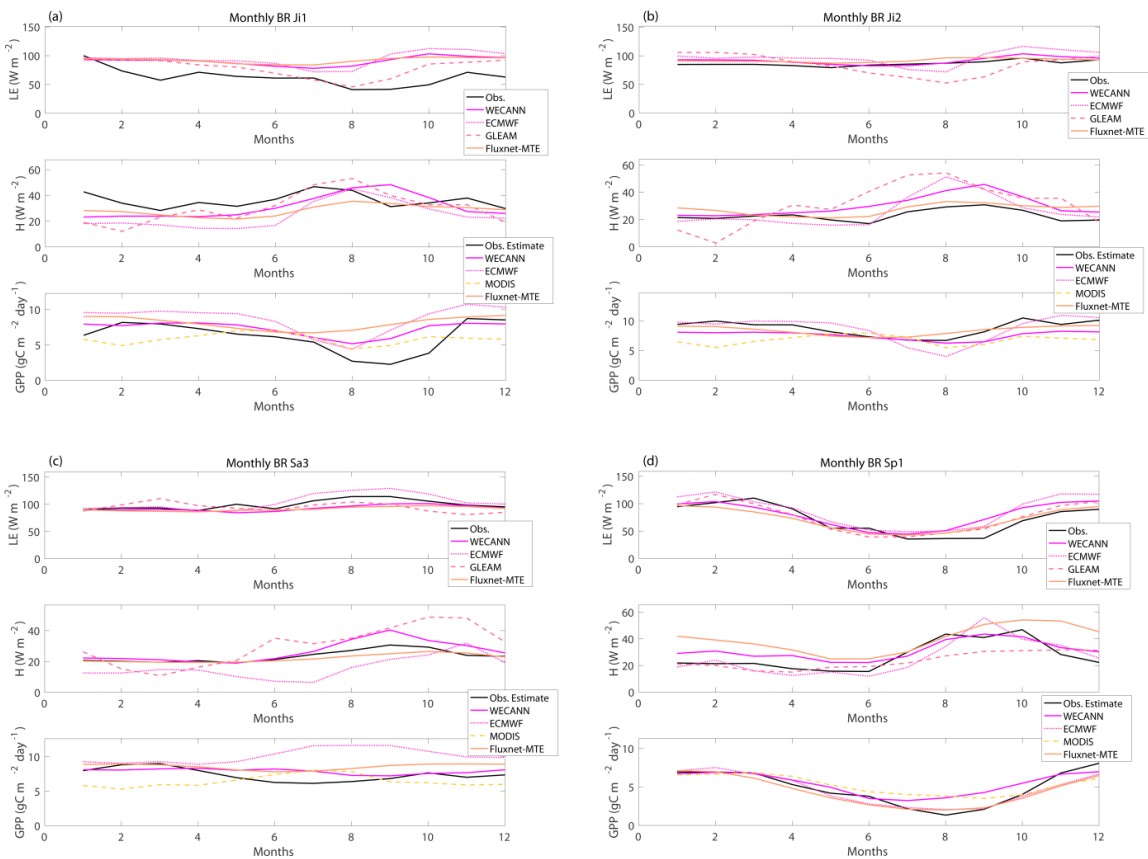

**Figure 12: Same as Figure 11 but for Brazilian sites (a) BR-Ji1, (b) BR-Ji2, (c) BR-Sa3, and (d) BR-Sp1.**

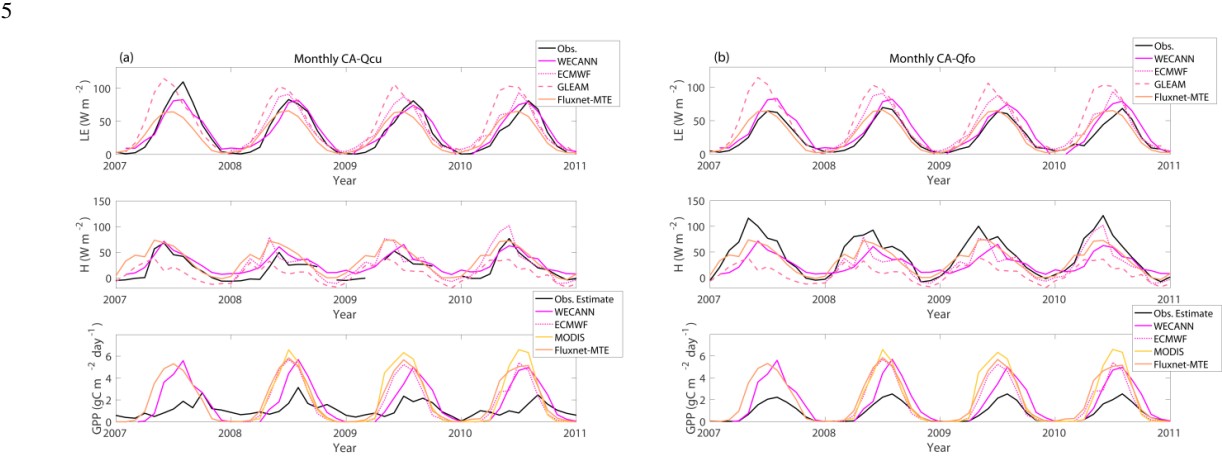

**Figure 13: Same as Figure 11 but for Canadian sites (a) CA-Qfo, and (b) CA-Qcu.**



**Figure 14: Same as Figure 11 but for US sites (a) US-ARM, (b) US-IB2, (c) US-ME2, (d) US-Ne1, and (e) US-SRG.**





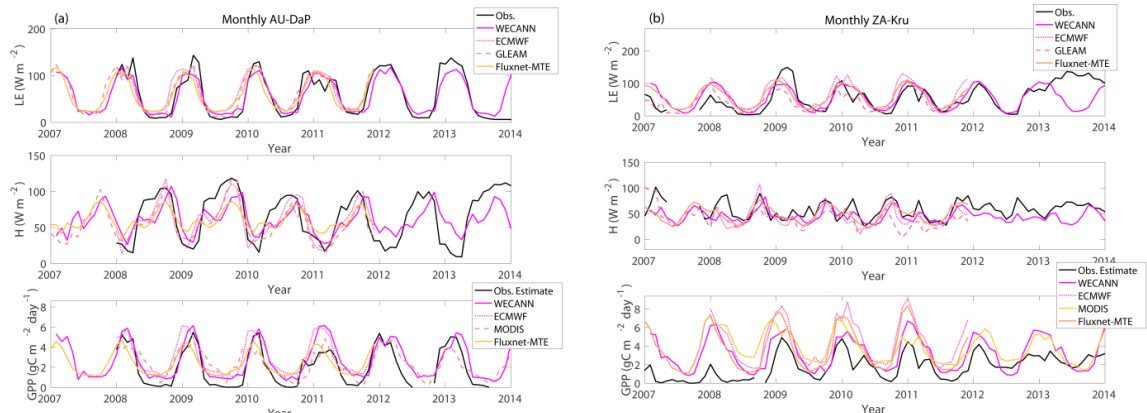

**Figure 15: Same as Figure 11 but for (a) AU-DaP, Australia, (b) ZA-Kru, South Africa**



**Table 1: Characteristics of products used for training of ANN**

| Product | Output variables used for training | Temporal Coverage | Spatial Coverage | Temporal Resolution | Spatial Resolution | Reference |
|---|---|---|---|---|---|---|
| GLEAM | LE, H | 1980 - 2015 | Global | Daily | $0.25° \times 0.25°$ | Martens et al., 2016 |
| ECMWF ERA HTESSEL | LE, H, GPP | 2008 - 2015 | Global | Daily | $0.25° \times 0.25°$ | Balsamo et al., 2009 |
| FLUXNET-MTE | LE, H, GPP | 1982 - 2012 | Global | Monthly | $0.5° \times 0.5°$ | Jung et al., 2009 |
| MODIS-GPP | GPP | 2000 - 2015 | Global | Monthly | $0.5° \times 0.5°$ | Running et al., 2004 |

5    **Table 2: Characteristics of observations used as input in the WECANN product**

| Variable | Product Name and Version | Temporal Coverage | Spatial Coverage | Temporal Resolution | Spatial Resolution | Reference |
|---|---|---|---|---|---|---|
| SIF | GOME-2 Fluorescence v26 | 2007-present | Global | Daily | $0.5° \times 0.5°$ | Joiner et al., 2013 |
| Net Radiation | CERES L3 SYN 1deg | 2002-present | Global | Monthly | $1° \times 1°$ | Wielicki et al., 1996 |
| Air Temperature | AIRS3STD v6.0 | 2002-present | Global | Daily | $1° \times 1°$ | Aumann et al., 2003 |
| Soil Moisture | ESA-CCI v2.3 | 1978-2015 | Global | Daily | $0.25° \times 0.25°$ | Liu et al., 2012 |
| Precipitation | GPCP 1DD v1.2 | 1996-2015 | Global | Daily | $1° \times 1°$ | Huffman et al., 2001 |
| Snow Water Equivalent | GLOBSNOW L3A v2 | 1979-present | Global | Daily | $25 \text{ km} \times 25 \text{ km}$ | Luojus et al., 2013 |

**Table 3: Comparison of WECANN retrievals with retrievals form an ANN without SIF as an input**

| | LE | | H | | GPP | |
|---|---|---|---|---|---|---|
| | RMSD [W m$^{-2}$] | $R^2$ | RMSD [W m$^{-2}$] | $R^2$ | RMSD [gC m$^{-2}$ day$^{-1}$] | $R^2$ |
| WECANN | 11.13 | 0.95 | 13.35 | 0.89 | 1.23 | 0.90 |
| ANN w/o SIF | 12.33 | 0.94 | 13.89 | 0.88 | 1.33 | 0.88 |



**Table 4: Statistics of LE retrievals compared to eddy-covariance measurements. Bold fonts represent best performing dataset statistics.**

| SiteID | Correlation Coefficient | | | | RMSE [W m$^{-2}$] | | | | Bias [W m$^{-2}$] | | | |
|---|---|---|---|---|---|---|---|---|---|---|---|---|
| | WECANN | FLUXNET-MTE | GLEAM | ECMWF | WECANN | FLUXNET-MTE | GLEAM | ECMWF | WECANN | FLUXNET-MTE | GLEAM | ECMWF |
| AT-Neu | 0.91 | 0.94 | 0.93 | **0.96** | 16.25 | 15.96 | 14.04 | **10.11** | -2.30 | -7.96 | **0.87** | **0.60** |
| AU-DaP | **0.89** | 0.80 | 0.87 | 0.82 | 22.25 | 26.85 | **21.42** | 25.42 | **-0.56** | 1.44 | 5.47 | 8.72 |
| BE-Bra | **0.96** | 0.78 | 0.80 | 0.79 | **8.58** | 18.20 | 20.89 | 19.86 | 11.28 | 14.02 | **9.46** | 23.34 |
| BR-Ji1 | 0.13 | 0.09 | **0.65** | 0.13 | 16.67 | 16.21 | **13.41** | 19.06 | 27.83 | 29.08 | **15.52** | 31.35 |
| BR-Ji2 | **0.77** | **0.77** | 0.04 | 0.49 | 4.16 | **2.93** | 18.88 | 11.24 | 4.85 | 5.32 | **-2.02** | 10.14 |
| BR-Sa3 | 0.70 | 0.73 | 0.24 | **0.91** | **6.52** | 6.74 | 10.87 | 7.84 | -6.61 | -7.85 | **-4.79** | 5.14 |
| BR-Sp1 | 0.84 | **0.90** | 0.92 | **0.90** | 14.74 | 12.50 | **11.10** | 12.70 | 7.63 | **-0.69** | 3.67 | 15.00 |
| CA-Gro | **0.94** | 0.88 | 0.79 | 0.87 | **11.61** | 14.45 | 22.27 | 17.40 | 9.96 | **3.20** | 17.79 | 13.94 |
| CA-Qcu | **0.96** | 0.84 | 0.78 | 0.89 | **9.58** | 15.67 | 23.77 | 13.89 | **1.64** | **-1.64** | 12.18 | 6.52 |
| CH-Fru | **0.81** | 0.76 | 0.69 | 0.66 | **21.35** | 24.09 | 27.85 | 29.96 | **1.63** | -3.52 | 4.96 | 11.22 |
| ES-LgS | **0.76** | 0.58 | -0.06 | 0.07 | **14.09** | 17.24 | 25.79 | 27.71 | **-1.21** | -9.17 | -19.21 | -4.45 |
| FI-Hyy | **0.97** | 0.90 | 0.82 | 0.88 | **7.05** | 12.21 | 18.52 | 16.16 | 1.01 | **-0.48** | 4.49 | 7.93 |
| US-ARM | 0.79 | **0.85** | **0.85** | 0.77 | 19.67 | **16.59** | 21.22 | 22.37 | 9.27 | **4.82** | 6.77 | 9.79 |
| US-IB2 | **0.95** | 0.86 | 0.86 | 0.84 | **19.31** | 26.15 | 26.79 | 28.50 | -11.49 | -17.56 | -16.27 | **2.71** |
| US-Me2 | **0.91** | 0.87 | 0.77 | 0.62 | **13.08** | 15.25 | 18.43 | 23.27 | -7.49 | -11.82 | **-3.11** | -7.61 |
| US-Ne1 | 0.85 | **0.90** | 0.85 | 0.65 | 35.71 | **25.59** | 31.03 | 45.20 | -28.02 | -15.06 | -23.50 | **-9.52** |
| US-SRG | **0.90** | 0.84 | 0.81 | 0.71 | **13.82** | 19.90 | 17.55 | 21.53 | -8.38 | -12.03 | **-4.91** | -13.81 |
| ZA-Kru | 0.50 | **0.69** | 0.59 | 0.65 | 38.81 | **28.79** | 30.22 | 31.82 | -5.09 | 14.70 | **-1.83** | 17.81 |
| **Average** | **0.81** | 0.78 | 0.68 | 0.70 | | | | | | | | |



**Table 5: Statistics of H retrievals compared to eddy-covariance measurements. Bold fonts represent best performing dataset statistics.**

| SiteID | Correlation Coefficient | | | | RMSE [W m⁻²] | | | | Bias [W m⁻²] | | | |
|---|---|---|---|---|---|---|---|---|---|---|---|---|
| | WECANN | FLUXNET-MTE | GLEAM | ECMWF | WECANN | FLUXNET-MTE | GLEAM | ECMWF | WECANN | FLUXNET-MTE | GLEAM | ECMWF |
| AT-Neu | **0.88** | 0.78 | 0.71 | 0.73 | **6.43** | 17.44 | 12.56 | 10.65 | 20.33 | 34.34 | **7.41** | 9.26 |
| AU-DaP | 0.68 | -0.85 | **0.82** | **0.82** | 25.68 | 63.64 | **18.93** | 18.91 | **-4.03** | -5.20 | -9.21 | -4.61 |
| BE-Bra | 0.87 | 0.78 | 0.78 | **0.90** | 18.01 | 18.97 | 22.51 | **15.06** | **-8.45** | 18.71 | -21.85 | -28.21 |
| BR-Ji1 | 0.29 | 0.42 | **0.57** | 0.45 | 9.27 | **5.67** | 10.09 | 9.27 | **-4.88** | -7.66 | -5.70 | -11.56 |
| BR-Ji2 | **0.83** | 0.64 | 0.51 | **0.85** | 4.79 | **3.57** | 14.10 | 8.09 | 7.00 | 4.28 | 7.92 | **3.02** |
| BR-Sa3 | **0.97** | 0.88 | 0.79 | 0.50 | 3.21 | **2.04** | 9.84 | 6.33 | 3.03 | **-1.34** | 6.84 | -7.45 |
| BR-Sp1 | **0.92** | 0.69 | 0.70 | 0.83 | **5.31** | 8.40 | 7.99 | 7.43 | 4.05 | 12.42 | -3.47 | **-1.50** |
| CA-Gro | 0.71 | 0.67 | 0.88 | **0.92** | 16.54 | 22.42 | 11.02 | **9.03** | -7.45 | **4.03** | -15.91 | -8.35 |
| CA-Qcu | **0.92** | 0.89 | 0.54 | 0.75 | **7.85** | 10.77 | 19.73 | 19.60 | 10.10 | 12.77 | **-8.76** | 10.87 |
| CH-Fru | **0.75** | 0.70 | 0.50 | 0.53 | **12.03** | 19.43 | 17.41 | 16.06 | 11.41 | 31.45 | 3.09 | **1.25** |
| ES-LgS | **0.87** | 0.59 | 0.75 | 0.81 | **26.93** | 46.40 | 34.49 | 32.83 | **0.47** | -23.91 | 4.07 | 17.25 |
| FI-Hyy | 0.85 | 0.91 | 0.86 | **0.93** | 16.51 | **12.76** | 16.67 | **12.45** | **0.01** | 5.78 | -9.44 | -14.33 |
| US-ARM | **0.80** | 0.52 | 0.59 | 0.70 | **18.47** | 28.31 | 21.24 | 22.11 | 1.44 | 14.82 | -11.65 | **1.16** |
| US-IB2 | 0.11 | -0.03 | **0.58** | 0.35 | 21.69 | 37.07 | **12.36** | 16.71 | 8.22 | 22.49 | **-5.68** | -5.97 |
| US-Me2 | 0.89 | 0.87 | 0.91 | **0.92** | 22.87 | 33.33 | **18.97** | 18.65 | -9.12 | -13.89 | -14.65 | 2.49 |
| US-Ne1 | 0.47 | -0.07 | **0.67** | 0.08 | 20.76 | 54.58 | **15.70** | 25.40 | 3.07 | 27.08 | -5.09 | **-1.80** |
| US-Ro1 | **0.81** | 0.63 | 0.42 | 0.33 | **10.00** | 27.64 | 16.87 | 18.35 | 9.54 | 27.42 | **-0.28** | -1.14 |
| US-Ses | **0.92** | 0.74 | 0.82 | **0.92** | **9.23** | 17.57 | 14.08 | 13.14 | -14.94 | -41.03 | -20.87 | **-6.25** |
| US-SRG | **0.87** | 0.02 | **0.86** | 0.88 | **13.98** | 30.95 | 14.80 | 18.01 | 6.53 | -25.52 | **0.81** | 28.58 |
| ZA-Kru | **0.59** | 0.18 | 0.47 | **0.59** | **12.97** | 32.57 | 20.12 | 17.27 | -12.78 | **-1.55** | -16.42 | -10.82 |
| **Average** | **0.75** | 0.50 | 0.69 | 0.69 | | | | | | | | |



10 **Table 6: Statistics of GPP retrievals compared to eddy-covariance measurements. Bold fonts represent best performing dataset statistics.**

| SiteID | Correlation Coefficient | | | | RMSE [gC m$^{-2}$ day$^{-1}$] | | | | Bias [gC m$^{-2}$ day$^{-1}$] | | | |
|---|---|---|---|---|---|---|---|---|---|---|---|---|
| | WECANN | FLUXNET-MTE | MODIS | ECMWF | WECANN | FLUXNET-MTE | MODIS | ECMWF | WECANN | FLUXNET-MTE | MODIS | ECMWF |
| AT-Neu | 0.86 | 0.92 | 0.92 | **0.94** | 1.35 | **1.05** | 1.08 | **1.04** | -0.52 | **0.14** | -0.38 | -0.93 |
| BE-Bra | **0.94** | 0.88 | 0.90 | 0.79 | **0.86** | 1.89 | 1.38 | 1.00 | **0.65** | 2.28 | 1.25 | **0.65** |
| BR-Ji1 | **0.80** | 0.51 | 0.34 | 0.76 | 1.53 | 1.92 | 2.09 | **1.47** | 1.14 | 1.92 | **-0.32** | 2.49 |
| BR-Ji2 | **0.84** | **0.83** | -0.03 | **0.82** | **0.82** | 0.79 | 1.57 | 1.34 | -1.20 | -0.40 | -1.98 | **-0.08** |
| BR-Sa3 | 0.44 | **0.70** | -0.83 | -0.73 | 0.85 | **0.71** | 1.74 | 1.86 | **0.53** | 1.20 | -0.92 | 2.77 |
| BR-Sp1 | **0.94** | 0.95 | 0.87 | **0.94** | 1.04 | **0.72** | 1.31 | **0.75** | 0.54 | -0.47 | 0.36 | **-0.20** |
| CA-Gro | **0.60** | 0.41 | 0.41 | 0.45 | **1.98** | 2.46 | 2.66 | 2.56 | **0.64** | 0.92 | 1.20 | 0.87 |
| CA-Qcu | **0.74** | 0.38 | 0.51 | 0.54 | **1.53** | 1.95 | 2.19 | 1.73 | 0.63 | 1.09 | 0.91 | **0.55** |
| CA-Qfo | **0.98** | 0.88 | 0.90 | 0.91 | **1.10** | 1.39 | 1.69 | 1.23 | 0.93 | 1.17 | 1.21 | **0.86** |
| CH-Fru | 0.91 | 0.91 | **0.93** | 0.90 | **1.13** | 1.63 | 1.51 | **1.12** | 0.70 | 1.61 | 0.97 | **0.60** |
| ES-LgS | **0.66** | 0.29 | -0.03 | -0.28 | 0.60 | **0.50** | 0.71 | 0.70 | 1.36 | **0.22** | 0.78 | **0.26** |
| FI-Hyy | **0.97** | 0.89 | 0.91 | 0.91 | **0.56** | 1.52 | 1.41 | 1.21 | **0.23** | 1.22 | 0.89 | 0.71 |
| US-IB2 | 0.85 | 0.86 | **0.96** | 0.84 | 1.42 | 1.94 | **0.80** | 1.27 | 1.41 | 0.98 | **0.57** | 0.97 |
| US-Me2 | 0.89 | 0.91 | **0.94** | 0.65 | 0.67 | **0.53** | 0.60 | 1.02 | -0.64 | **-0.08** | -0.23 | -1.22 |
| US-Ne1 | 0.77 | **0.88** | 0.69 | 0.54 | 2.33 | **1.69** | 2.43 | 3.04 | 0.80 | 0.30 | **-0.03** | -0.42 |
| US-Ses | 0.63 | **0.76** | 0.42 | 0.58 | 0.53 | **0.40** | 0.62 | 0.60 | 0.38 | **0.03** | 0.15 | -0.51 |
| US-SRG | 0.76 | **0.87** | 0.52 | 0.59 | 0.51 | **0.38** | 0.59 | 0.59 | 0.55 | 0.50 | 0.23 | **-0.07** |
| ZA-Kru | **0.69** | 0.67 | 0.40 | **0.70** | **1.31** | 1.61 | 1.58 | 1.66 | **1.27** | 2.33 | 1.87 | 2.93 |
| **Average** | **0.80** | 0.75 | 0.54 | 0.60 | | | | | | | | |