# Peer review of "Water, Energy, and Carbon with Artificial Neural Networks (WECANN): A statistically-based estimate of global surface turbulent fluxes using solar-induced fluorescence"

_Biogeosciences, 2016_

## Referee Comment (RC1) · Anonymous Referee #1 · 20 Dec 2016

Review on BG-2016-495 General comments: The authors proposed a new global product of GPP, ET and H by using ANN. The manuscript is well written and the topic falls on to the scope of the journal. I do have several concerns.

First, the authors highlighted the use of SiF as input data. I see SiF was the only input data related to vegetation. Therefore, with/without SiF in WECANN must give different flux estimates. What happens if the authors use EVI or NDVI instead of SiF? Any significant difference in WECANN performance?

[Figure]

Second, what is the significant contribution from this work? Spatial (1 degree) and temporal (monthly) resolutions are too coarse. The approach is on the similar family of other machine learning methods (e.g. see Tramontana et al 2016 Biogeosciences). Stress the novelty of this manuscript. If there is any new discovery, then highlight it.

Tramontana, G., Jung, M., Schwalm, C.R., Ichii, K., Camps-Valls, G., Ráduly, B., Reichstein, M., Arain, M.A., Cescatti, A., Kiely, G., Merbold, L., Serrano-Ortiz, P., Sickert, S., Wolf, S., Papale, D. (2016) Predicting carbon dioxide and energy fluxes across global FLUXNET sites with regression algorithms. Biogeosciences 13, 4291-4313.

Third, the authors used MPI-BGC product as a training dataset while testing the product against FLUXNET data. As MPI-BGC product was trained against FLUXNET dataset, the approach is self-correlated. Why not evaluating the product against independent datasets from MPI-BGC? E.g. water balance derived ET in basin scale.

Fourth, the spatial domain should be clearly defined. The authors said it is global product, but it did not include Antarctica and Greenland. Given the coarse resolution (100 km), most islands are likely uncovered but the global map (Fig 2) showed fluxes in some islands. How did it happen? Also, how to treat with water fraction for each 1-degree pixel?

Fifth, I recommend showing global uncertainty maps for GPP, LE, H. I think one of strengths in WECANN is its ability to quantify uncertainty. Show the uncertainty map and discuss where and why uncertainties are high. Also quantify uncertainties in global values (e.g. XXX PgC yr-1 +- Y PgC yr-1).

Sixth, test global more carefully. When I look at Fig 2, I found higher ET in mid to south east South America (e.g. cerrado) compared to other global ET products. Also, your ET in this region is relatively very high compared to your GPP map. So, water use efficiency will be very low in this region, which is unlikely. See global distribution of C4 maps. Higher proportion in C4 in this area is likely to lead higher water use efficiency. It is notable that your ANN did not consider C4 information.

Specific comments: P6: why only 21 FLUXNET sites were used? More than 150 sites data are open to public

P6 L23-24: The authors explained that target data is used for training, validation, and testing. I am confused with the terminology of validation and testing. How do they differ? Also, in L36, "after training, ….. was evaluated". Here, does "evaluation" indicate validation or testing? I recommend clearly defining each term, and use them consistently across the whole manuscript.

P6 L30: NN -> ANN (?)

P7 L9: Please define "multiple datasets." Is this training dataset?

P7 L12: What is "this" in "this prior distribution"?

P8 L20: Is this "target estimate" from 3.2?

P8 L22: Add another unit for GPP as PgC yr-1, which could be easily compared to the other studies. Same for LE (km3).

P9 L29: I was surprised to see the reduction of GPP in the Saharan Desert after removing SiF. How to interpret this as we know there must be zero GPP? Also, exclusion of SiF in LE made mixed tendencies in this region. As we are confident LE and GPP are close to nil in this area, it will be interesting to test the impacts of inclusion/exclusion in SiF on LE and GPP here.

P10 L3: All three R2 looks too similar, so it is hard to tell 0.96 is higher than 0.94.

P10: The authors compared WECANN to FLUXNET-MTE, ECMWF, GLEAM and MODIS-GPP which were the training data for WECANN. I feel there should be self-correlation, so I am curious whether this is a reasonable approach.

P10 L8: I know there are few eddy flux tower data in India, so FLUXNET-MTE might involve higher uncertainty. However, this is the same situation for WECANN as it used FLUXNET-MTE and others, which are all uncertain as training dataset.

P10 L4: Be quantitative. Report bias.

P10 L20: Define "G"

P11: Many contents in this page should move to Methods.

P12 L5-6: Then why not removing this site given obvious deficiencies?

P12- : As the authors well recognized, I feel it is odd to compare 1 degree WECANN to several hundred meters in flux towers. All discussion from this comparison seems too subjective. I think "validation" of 0.5-degree product is unlikely possible. As your products are too coarse, I would recommend evaluating at larger scales. For example, look interannual variability of global GPP (PgC yr-1), ET (W m-2), and H (W m-2) and compare to atmospheric inversion estimates. Test whether your product could capture big climate extreme events such as Russian heatwave, Texas drought etc. Compare to other existing global land surface products which were not used as input/training dataset in WECANN.

---

## Referee Comment (RC2) · Anonymous Referee #2 · 2 Jan 2017

This manuscript is well written and deserves consideration for publication in this journal. However, I have the following issues that need to be addressed.

The paper proposes an empirical machine learning 'meta-model' to try to learn from different existing datasets to combine their strengths and factor out their limitations. On one hand, I appreciate this effort to bring together different datastreams and somehow harmonize them through this new consolidated product, but on the other, I am wary of this approach of blindly adding further algorithmic layers without really trying to un-

derstand mechanistically why the initial datasets have shortcomings. If all products are equally off in some parts, combining them just gives the false impression we are going in the right direction while reality is still off. Also, the FLUXNET-MTE used as training is already a machine learning product driven by various input variables, very much like WECANN is. Furthermore, there is quite some circularity in the work since the FLUXNET-MTE and MODIS GPP are both strongly based on the same fluxtowers used here for validation. I deem that all these points need to be acknowledged clearly and discussed thoroughly.

Could you specify why you use the SYN products (Level 3) from CERES instead of the EBAF ones (Level 3B)? The later have been energy balanced according to the product specifications. Wouldn't this be an advantage in your case?

In the construction of the ANN, I would welcome to have some justification of why tangent sigmoid transfer functions are used instead of linear ones. I know this is often done, but it seems very arbitrary. Also, I did not quite understand how the 20% of 'testing' data is used. I clearly see that 60% are used for training and 20% for validation, but how exactly do you use the other 20%? Perhaps this just needs some rephrasing in the text for clarification.

Comparison with fluxtower measurements is not appropriate as the difference in spatial support is just too different (1 squared degree vs <1km2). Saying that WECANN performs better that other products based on individual towers while all these products cover such a larger area (by several orders of magnitude) just does not make much sense (even if it has been done in other studies). The authors would need to do some filtering of the towers to select only those that can be considered representative (e.g. http://doi.org/10.1016/j.rse.2016.04.027), although I doubt this would leave many valid towers for pixels of 1 squared degree. Another option that may be more feasible would be to make an evaluation at a larger aggregation scale, such as for clusters of similar climates and plant functional types. Making such averages from the fluxtowers on one side and from all pixels that are comparable in this respect on the other would reduce

the number of measurements for validation, but would render them more credible. I would also suggest to exploit more of the available towers in the Fluxnet2015 dataset instead of only 21.

The part pretending to demonstrate the value of SIF is also inadequate as the authors only test the effect of removing this one input. By doing so, any information of the actual vegetation phenology is lost, which would necessarily reduce the performance. What would be interesting would be to show that SIF provides better information that the classical vegetation indices like NDVI or EVI. To do so, the SIF input of the ANN should be replaced by one of these and then a judgement on the pertinence of SIF can be made.

Finally, the manuscript is often too long and too descriptive in several parts describing the graphs and maps. This needs to be reduced drastically. Most of what is being said can be easily inferred from the reader by looking at the graphs, while deeper discussion on why discrepancies occur between products and fluxtowers would be more welcome. Also, please remove the extensive references to different parts of the text and the description of the structure of the paper (e.g. page 3 lines 10-20), I think they are lengthening the text needlessly.

---

## Author Comment (AC1) · 11 Feb 2017

**The authors proposed a new global product of GPP, ET and H by using ANN. The manuscript is well written and the topic falls on to the scope of the journal. I do have several concerns.**

*Re:* We thank the referee for his/her positive comments.

**First, the authors highlighted the use of SiF as input data. I see SiF was the**

**only input data related to vegetation. Therefore, with/without SiF in WECANN must give different flux estimates. What happens if the authors use EVI or NDVI instead of SiF? Any significant difference in WECANN performance?**

*Re:* We agree with the referee's point on the evaluation of no-SIF retrieval. Therefore, in the revised manuscript we included comparisons with an artificial neural networks retrieval that has either NDVI or EVI as input instead of SIF. We comment on the differences and similarities, and why SIF is a better input for this retrieval, in particular highlighting the differences in terms of vegetation structure impact on SIF and the impact of saturation of vegetation indices (especially in forested areas and agricultural regions).

**Second, what is the significant contribution from this work? Spatial (1 degree) and temporal (monthly) resolutions are too coarse. The approach is on the similar family of other machine learning methods (e.g. see Tramontana et al 2016 Biogeosciences). Stress the novelty of this manuscript. If there is any new discovery, then highlight it.**

*Re:* There are two major new contributions in this study:

1. Using remotely-sensed SIF to estimate surface fluxes.

2. Using a machine learning algorithm (in this case artificial neural networks) to estimate fluxes from remote sensing observations at global scale.

The Tramontana et al 2016 paper uses a regression model to upscale fluxes from FLUXNET observations. However, we use remote sensing observations to estimate fluxes, and use FLUXNET towers to evaluate the performance of our retrievals. Therefore the strategy is pretty different compared to the Tramontana et al retrieval. In addition, our main objective is to show that SIF provides useful information on the rates of photosynthesis and evapotranspiration. To our knowledge this is the first direct estimate of fluxes based on SIF data. We revised portions of the text in the introduction section to make sure the novelty of our approach is clearly stated.

**Third, the authors used MPI-BGC product as a training dataset while testing the product against FLUXNET data. As MPI-BGC product was trained against FLUXNET dataset, the approach is self-correlated. Why not evaluating the product against independent datasets from MPI-BGC? E.g. water balance derived ET in basin scale.**

*Re:* That is not exactly correct. We train our algorithm against a target dataset which is derived from three products (including MPI-BGC) by using the Triple Collocation method and assigning *a priori* weights to every product in each pixel. This means that our target dataset has collective information from all three products and not just MPI-BGC. Therefore, we acknowledge that there is some information carried from FLUXNET tower data in MPI-BGC to our training process. However, the degree of self-correlation is mainly true for the comparison of FLUXNET-MTE with the tower estimates but not for the WECANN estimate. Indeed, it has been shown (see Jimenez et al. 2009 for instance) that the spatial and temporal correlations of a global artificial neural network are not due to the initial training dataset but to the remote sensing observations used as input.

Jimenez, C., Prigent, C., & Aires, F. (2009). Toward an estimation of global land surface heat fluxes from multisatellite observations. *Journal of Geophysical Research-Atmospheres, 114*(D6), D06305.

Moreover, given that FLUXNET-MTE uses flux tower estimates for the retrieval we would have expected this product to be the better one when compared to local eddy covariance data. In fact, we show the opposite. WECANN informed by direct remote sensing observations typically outperforms FLUXNET-MTE, especially in terms of seasonal cycle, further emphasizing the information content provided by remote sensing data.

Finally, conducting a water balance analysis will be informative while it has its own challenges because of uncertainties in other inputs for the water balance, in order to close the budget as multiple sources of information needs to be used. However, we believe that this is beyond the scope of the current study which is solely focused on developing the retrieval algorithm, also the other referee commented on the length of the paper asking us to reduce it.

**Fourth, the spatial domain should be clearly defined. The authors said it is global product, but it did not include Antarctica and Greenland. Given the coarse resolution (100 km), most islands are likely uncovered but the global map (Fig 2) showed fluxes in some islands. How did it happen? Also, how to treat with water fraction for each 1-degree pixel?**

*Re:* Thank you for the comments. We have now revised the description in the introduction section to clearly note what the coverage of the new product is.

**Fifth, I recommend showing global uncertainty maps for GPP, LE, H. I think one of strengths in WECANN is its ability to quantify uncertainty. Show the uncertainty map and discuss where and why uncertainties are high. Also quantify uncertainties in global values (e.g. XXX PgC yr-1 +- Y PgC yr-1).**

*Re:* In the revised manuscript we now include uncertainty estimates based on errors in the input data propagated into the network. We report a global average value as error is spatially and temporally variable.

**Sixth, test global more carefully. When I look at Fig 2, I found higher ET in mid to south east South America (e.g. cerrado) compared to other global ET products. Also, your ET in this region is relatively very high compared to your GPP map. So, water use efficiency will be very low in this region, which is unlikely. See global distribution of C4 maps. Higher proportion in C4 in this area is likely to lead higher water use efficiency. It is notable that your ANN did not consider C4 information.**

*Re:* The referee's point is an important one. The SIF relationship with GPP will likely change in C4 plants. However, we explicitly did not want to impose the C4/C3 (or even CAM) delimitation in the artificial neural network as it would be highly dependent on the quality of the classification map used and might be time varying. Given that we do not have partitioning of transpiration to total ET, it is hard to say whether the water use efficiency is indeed low or if rain re-evaporation and soil evaporation is the main process explaining the difference. We have nonetheless added a comment in the text emphasizing the referee's point(s).

**Specific comments:**

**P6: why only 21 FLUXNET sites were used? More than 150 sites data are open to public**

*Re:* We had selected this 21 sites to represent a range of climatic conditions along a geographical gradient for validation of our retrieval. Presenting evaluation metrics and temporal time series for 150 sites would lengthen the manuscript and make it hard to read. However, in the revised manuscript we will present summary statistics from comparison of WECANN retrievals against a much larger number of tower data from the FLUXNET 2015 dataset in the Appendix.

**P6 L23-24: The authors explained that target data is used for training, validation, and testing. I am confused with the terminology of validation and testing. How do they differ? Also, in L36, "after training, . . ... was evaluated". Here, does "evaluation" indicate validation or testing? I recommend clearly defining each term, and use them consistently across the whole manuscript.**

*Re:* We apologize for the confusion. The training, validation and testing proportions are related to the training phase of the retrieval. The back propagation algorithm uses a portion of the training data for training (basically estimating the weights of each neuron), and other portions of the training data for validation and testing that aims at checking the convergence of the training step. While after the training is done, we use a subset

of data that were not used in the training process for evaluation. We revised the text in the new version of the manuscript to clarify these terminologies.

**P6 L30: NN ->ANN (?)**

*Re:* Our apologies, this has been corrected in the revised manuscript.

**P7 L9: Please define "multiple datasets." Is this training dataset?**

*Re:* This refers to the three products that we use (together with error weights from Triple Collocation) to define a target dataset for training. We revised the text in the new version of the manuscript to clarify this.

**P7 L12: What is "this" in "this prior distribution"?**

*Re:* It refers to the pseudo Bayesian training mentioned in the lines before. We revised the text in the new version and clarified the point.

**P8 L20: Is this "target estimate" from 3.2?**

*Re:* Yes, this is the same. We made changes to section 3 and 4 of the manuscript in the new version to clarify all these terminologies.

**P8 L22: Add another unit for GPP as PgC yr-1, which could be easily compared to the other studies. Same for LE (km3).**

*Re:* Thanks for noting this. We included the new units along with previous ones in the new version of the manuscript.

**P9 L29: I was surprised to see the reduction of GPP in the Saharan Desert after removing SiF. How to interpret this as we know there must be zero GPP? Also, exclusion of SiF in LE made mixed tendencies in this region. As we are confident LE and GPP are close to nil in this area, it will be interesting to test the impacts of inclusion/exclusion in SiF on LE and GPP here.**

*Re:* This observation is true, and is caused by noise. As noted correctly by the referee,

LE and GPP are close to zero in this region; therefore, the difference between the two retrievals (with and without SIF) divided by the small amount of flux in this region is on the order of noise in the retrievals. While the percentages of change are notable, the absolute values of difference between the two retrievals are less than 2 w m$^{-2}$ for LE and less than 0.7 gC m$^{-2}$ day$^{-1}$ for GPP. In addition, the noisy pattern does not show up in the H retrievals in this region. This is also another sign that the change patterns in LE and GPP are due to noise. We now emphasize this point raised by the referee in the newer version of the manuscript.

**P10 L3: All three R2 looks too similar, so it is hard to tell 0.96 is higher than 0.94.**

*Re:* Yes, we agree and have typically highlighted with bold fonts all comparable products for a fair comparison. This is further emphasized in the text of the revised version of the manuscript.

**P10: The authors compared WECANN to FLUXNET-MTE, ECMWF, GLEAM and MODIS-GPP which were the training data for WECANN. I feel there should be self-correlation, so I am curious whether this is a reasonable approach.**

*Re:* We feel that there is a misconception here. We did not compare WECANN to these products. Indeed, we are comparing WECANN and the other products against FLUXNET individual tower observations and reporting the performance of each one of them. We show that WECANN has a better performance while it is trained on the collective set of data from these products, and we believe this is an advantage of our approach.

**P10 L8: I know there are few eddy flux tower data in India, so FLUXNET-MTE might involve higher uncertainty. However, this is the same situation for WE-CANN as it used FLUXNET-MTE and others, which are all uncertain as training dataset.**

*Re:* It is true that there are few towers in India, but our retrieval does not rely on regional

towers to estimate surface fluxes. We train an artificial neural network algorithm using the three training products that is mentioned in the manuscript for all the pixels. That means we use the information from all the pixels over the globe to train one retrieval algorithm. This algorithm uses remote sensing observation at each point in time and space to retrieve surface fluxes. Therefore, lack of FLUXNET towers in any part of the globe would not impact the accuracy of WECANN retrievals, while this can be an issue for products that upscale tower-based observations to estimate fluxes across the globe.

**P10 L4: Be quantitative. Report bias.**

*Re:* Thanks for noting this. The point we have raised in this line (on the spread of scatter plots) can be quantitatively compared using the RMSD value that are provided in the figures. In the revised manuscript, we report this in the text as well to support the argument we are making.

**P10 L20: Define "G"**

*Re:* Corrected in the revised manuscript.

**P11: Many contents in this page should move to Methods.**

*Re:* We agree that some of the information on the description of FLUXNET data used here might be relevant to section 2 on "Data". In the revised manuscript, we re-organized the text and moved these contents to section 2.3.

**P12 L5-6: Then why not removing this site given obvious deficiencies?**

*Re:* We believe that it is informative to include this comparison, and show that the representativeness area can be a challenge in comparing large-scale remote sensing based retrievals to point based tower data. In this case, we have good knowledge of the site and its surrounding region so it is possible to investigate if the tower data is representative of the larger scale fluxes.

**P12- : As the authors well recognized, I feel it is odd to compare 1 degree WE-CANN to several hundred meters in flux towers. All discussion from this comparison seems too subjective. I think "validation" of 0.5-degree product is unlikely possible. As your products are too coarse, I would recommend evaluating at larger scales. For example, look interannual variability of global GPP (PgC yr-1), ET (W m-2), and H (W m-2) and compare to atmospheric inversion estimates. Test whether your product could capture big climate extreme events such as Russian heatwave, Texas drought etc. Compare to other existing global land surface products which were not used as input/training dataset in WECANN.**

*Re:* We would like to emphasize that any new retrieval algorithm development requires some validation against ground truth observations. In fact, other reviewers wanted to see some comparison. While there is some caveat in validation against point based tower data, these are the only ground based observations available for such a validation. Moreover, in the comparison against tower data many large scale variabilities such as seasonal cycle are comparable to pixel based retrievals. This is also the case for interannual variability, and we have discussed them in detail, in section 4.4 of the original manuscript. For instance, the phenology has a strong impact on the seasonal cycle of the fluxes and is here clearly highlighted when comparing the different products to flux tower estimates.

In the revised manuscript, we highlighted this limitation in section 4.4, while noting that comparison against ground-based tower observations is common practice and is what the community indeed looks for when a new retrieval algorithm is developed. We believe that specific drought or flood events would lack the generality provided here when comparing all years/months.

---

## Author Comment (AC2) · 11 Feb 2017

**This manuscript is well written and deserves consideration for publication in this journal. However, I have the following issues that need to be addressed.**

*Re:* We appreciate the referee's positive feedback and provide responses to his/her comments below.

**The paper proposes an empirical machine learning 'meta-model' to try to learn**

**from different existing datasets to combine their strengths and factor out their limitations. On one hand, I appreciate this effort to bring together different datastreams and somehow harmonize them through this new consolidated product, but on the other, I am wary of this approach of blindly adding further algorithmic layers without really trying to understand mechanistically why the initial datasets have shortcomings. If all products are equally off in some parts, combining them just gives the false impression we are going in the right direction while reality is still off. Also, the FLUXNET-MTE used as training is already a machine learning product driven by various input variables, very much like WECANN is. Furthermore, there is quite some circularity in the work since the FLUXNET-MTE and MODIS GPP are both strongly based on the same fluxtowers used here for validation. I deem that all these points need to be acknowledged clearly and discussed thoroughly.**

*Re:* We acknowledge this concern, and would like to bring the following points to the referee's attention:

1. The WECANN machine learning retrieval is quite different than FLUXNET-MTE in the sense that we use remote sensing observations and estimate surface turbulent fluxes while FLUXNET-MTE upscales tower-based observations to estimate surface fluxes at global scale. Although both approaches use machine learning techniques (artificial neural networks in the case of WECANN and regression in the case of FLUXNET-MTE) their retrieval algorithms are quite different and directly informed by only remote sensing observations in WECANN, which we believe is an important means of better constraining the retrievals.

2. Moreover, our training approach uses all the spatial and temporal observations during the training period (2008-2010) to develop one single neural network for the global retrievals. This network is then used with remote sensing observations as input to retrieve surface fluxes. Therefore, if in a few percentage of times and

pixels, all the three training products are equally off this will be mitigated by the larger number of pixel/time data points that have more accurate estimates in other places and other times. In addition, the network can even correct the seasonal cycle when learning from an incorrect seasonal cycle training data, as the remote sensing inputs provide the information on the seasonal cycle directly. This has already been demonstrated by previous studies such as Jimenez et al. (2009).

Jimenez, C., Prigent, C., Aires, F. (2009). Toward an estimation of global land surface heat fluxes from multisatellite observations. *Journal of Geophysical Research-Atmospheres, 114*(D6), D06305.

3. On the issue of validation against FLUXNET tower data, we acknowledge that two of the training products use FLUXNET data for their calibration or as input but virtually all products have been calibrated in some ways or tested against eddy-flux tower. It does not however mean that the products are not independent: indeed the products are typically calibrated to reproduce either the annual mean or are adjusted per season at very few sites but not the exact temporal structures of the eddy-covariance observations (except for FLUXNET-MTE). Here, we use the three training products together with *a priori* weights calculated from Triple Collocation to define a target dataset that has collective information from all three of them. Then, we train our network on the target dataset. Finally, we validate the retrievals of WECANN against FLUXNET tower data and compare its performance with the performance of the three training products. While some information from FLUXNET observations propagates through the training products to WECANN training, the comparison results against FLUXNET observations show that WECANN learns from the three products collectively and performs better than any of them individually, emphasizing that our strategy works well. In addition, it is clear that WECANN does not have the seasonal biases seen in most retrievals (see e.g. FI-Hyy site where WECANN correctly captures this cold region's photosynthesis and evapotranspiration compared to the other products).

We also made changes to sections 3 and 4 of the manuscript in the revised version to better reflect on these points.

**Could you specify why you use the SYN products (Level 3) from CERES instead of the EBAF ones (Level 3B)? The later have been energy balanced according to the product specifications. Wouldn't this be an advantage in your case?**

*Re:* Our goal here, as also mentioned in section 1 of the manuscript, is to only use remotely-sensed observations as input. The EBAF product is based on a model with some remote sensing observations; therefore, we decided to use the SYN product to avoid any model addition.

**In the construction of the ANN, I would welcome to have some justification of why tangent sigmoid transfer functions are used instead of linear ones. I know this is often done, but it seems very arbitrary.**

*Re:* In this case, we tried the tangent sigmoid (the common choice) as well as linear, and did not see any notable changes in the performance of the network. Therefore, we chose to use the typical tangent sigmoid function. This has been added to the text .

**Also, I did not quite understand how the 20% of 'testing' data is used. I clearly see that 60% are used for training and 20% for validation, but how exactly do you use the other 20%? Perhaps this just needs some rephrasing in the text for clarification.**

*Re:* We apologize for the confusion on this matter. In the revised manuscript, we explained this in more detail. In summary, these percentages are for the data that are used in the training process. This is standard practice in artificial neural networks training. The back-propagation algorithm uses the training portion of the data for estimating the weights of neurons in the network, and the validation and test data are used to evaluate convergence of the training. These are separate than the data that we used for validation later on. Our validation, uses a subset of data that are not used in the

training, to make sure the network is not over-fitted to the training data.

**Comparison with fluxtower measurements is not appropriate as the difference in spatial support is just too different (1 squared degree vs <1km2). Saying that WECANN performs better that other products based on individual towers while all these products cover such a larger area (by several orders of magnitude) just does not make much sense (even if it has been done in other studies). The authors would need to do some filtering of the towers to select only those that can be considered representative (e.g. http://doi.org/10.1016/j.rse.2016.04.027), although I doubt this would leave many valid towers for pixels of 1 squared degree. Another option that may be more feasible would be to make an evaluation at a larger aggregation scale, such as for clusters of similar climates and plant functional types. Making such averages from the fluxtowers on one side and from all pixels that are comparable in this respect on the other would reduce the number of measurements for validation, but would render them more credible. I would also suggest to exploit more of the available towers in the Fluxnet2015 dataset instead of only 21.**

*Re:* We acknowledge that comparison against point based tower data has its own limitation (as we also have noted in the manuscript), but these are the only ground-based validation data that is available for evaluating a new global product. For this reason, we used a selection of sites spanning a geographical gradient and provided detail explanation on the comparison results in each site based on the knowledge of the land cover / land use around the site to make sure the differences, if any, can be explained either by uncertainties in WECANN retrieval or representativeness of the of the towers. As the referee notes, filtering the towers based on representativeness might not leave us with any tower to use.

In the original manuscript, we only used 21 towers that were selected to represent a wide range of climatic conditions and we would be able to explain the results of each one of them, including the time series plots in detail. However, due to the request of

both referees we will include summary statistics from comparison of WECANN against all the sites in FLUXNET 2015. We also want to empathize that some features of the flux tower observations such as phenology and seasonality are correctly picked up by our retrieval compared to other products and are only moderately affected by the heterogeneity within the pixels (except if there would be a very different even composition of deciduous and conifers for instance). We also want to emphasize that this is the first retrieval using SIF and directly assessing its usefulness for flux retrievals. For those reasons, we believe that using coarse resolution algorithms does not alter the interest of the study.

**The part pretending to demonstrate the value of SIF is also inadequate as the authors only test the effect of removing this one input. By doing so, any information of the actual vegetation phenology is lost, which would necessarily reduce the performance. What would be interesting would be to show that SIF provides better information that the classical vegetation indices like NDVI or EVI. To do so, the SIF input of the ANN should be replaced by one of these and then a judgement on the pertinence of SIF can be made.**

*Re:* We appreciate referee's comment on this point. In order to better demonstrate the value of SIF observations we have included comparisons with retrievals that have only NDVI or EVI instead of SIF in the revised manuscript. This better shows the value of having SIF as an input in retrieving surface fluxes. Thank you for this important comment. The results further emphasize the difference between SIF and purely vegetation structure and phenology (as well as saturation effects of vegetation indices).

**Finally, the manuscript is often too long and too descriptive in several parts describing the graphs and maps. This needs to be reduced drastically. Most of what is being said can be easily inferred from the reader by looking at the graphs, while deeper discussion on why discrepancies occur between products and fluxtowers would be more welcome. Also, please remove the extensive references to different parts of the text and the description of the structure of the**

**paper (e.g. page 3 lines 10-20), I think they are lengthening the text needlessly.**

*Re:* Given the novelty of the approach we feel that it is important to correctly describe the different steps of the analysis as many are relatively new such as the machine learning and the triple collocation. We had received the opposite comments before that we were not sufficiently describing the details; hence, why the article goes into the details of the retrieval.

---

## Author Response (AR1)

Response to Referee 1

**Water, Energy, and Carbon with Artificial Neural Networks (WECANN): A statistically-based estimate of global surface turbulent fluxes using solar-induced fluorescence (Manuscript # bg-2016-495)**

| Comments | Responses/Actions |
|---|---|
| The authors proposed a new global product of GPP, ET and H by using ANN. The manuscript is well written and the topic falls on to the scope of the journal. I do have several concerns. | We thank the referee for his/her positive comments. |
| First, the authors highlighted the use of SiF as input data. I see SiF was the only input data related to vegetation. Therefore, with/without SiF in WECANN must give different flux estimates. What happens if the authors use EVI or NDVI instead of SiF? Any significant difference in WECANN performance? | We agree with the referee's point on the evaluation of no-SIF retrieval. Therefore, in the revised manuscript we included comparisons with an artificial neural networks retrieval that has either NDVI or EVI as input instead of SIF. We comment on the differences and similarities, and why SIF is a better input for this retrieval, in particular highlighting the differences in terms of vegetation structure impact on SIF and the impact of saturation of vegetation indices (especially in forested areas and agricultural regions). Results are summarized in section 4.5 and Tables S1-S3 in the supplementary materials. |
| Second, what is the significant contribution from this work? Spatial (1 degree) and temporal (monthly) resolutions are too coarse. The approach is on the similar family of other machine learning methods (e.g. see Tramontana et al 2016 Biogeosciences). Stress the novelty of this manuscript. If there is any new discovery, then highlight it. | There are two major new contributions in this study:
 1- Using remotely-sensed SIF to estimate surface fluxes.
 2- Using a machine learning algorithm (in this case artificial neural networks) to estimate fluxes from remote sensing observations at global scale.
 The Tramontana et al 2016 paper uses a regression model to upscale fluxes from FLUXNET observations. However, we use remote sensing observations to estimate fluxes rather than relying on the representativeness of spatially limited FLUXNET eddy-covariance data like the Tramontana et al approach and its predecessors. Most importantly we use l SIF as an indicator of vegetation activity. Therefore, the strategy is pretty different compared to the Tramontana et al retrieval. In addition, our main objective is to show that SIF provides useful information on the rates of photosynthesis and evapotranspiration. To our knowledge, this is the first direct estimate of fluxes based on SIF data. We revised portions of the text in the introduction section (Page 2, Lines 30-33) to make sure the novelty of our approach is clearly stated. |

| | |
|---|---|
| Third, the authors used MPI-BGC product as a training dataset while testing the product against FLUXNET data. As MPI-BGC product was trained against FLUXNET dataset, the approach is self-correlated. Why not evaluating the product against independent datasets from MPI-BGC? E.g. water balance derived ET in basin scale. | We do not share the referee's perspective. While we acknowledge that there is some information carried from FLUXNET tower data into the FLUXNET-MTE dataset and therefore in some part of the training data, we believe this cross-correlation is likely to be small. We train our algorithm against a target dataset which is derived from three products (including MPI-BGC) by using the Triple Collocation method and assigning *a priori* weights to every product in each pixel. This means that our target dataset has collective information from all three products and not just MPI-BGC. Furthermore, the correlation between FLUXNET and FLUXNET-MTE data is also imperfect (cfg. Figure 2 in Tramontana et al 2016). Interestingly, WECANN typically outperforms other products (including FLUXNET-ME, which is expected to have a stronger correlation to FLUXNET data) especially in terms of seasonal cycle. This further emphasizes the information content provided by remote sensing data which are used as additional inputs This is consistent with previous work (see Jimenez et al. 2009 for instance) that the spatial and temporal correlations of a global artificial neural network are not due to the initial training dataset but to the remote sensing observations used as input. Jimenez, C., Prigent, C., Aires, F. (2009). Toward an estimation of global land surface heat fluxes from multi-satellite observations. Journal of Geophysical Research-Atmospheres, 114(D6), D06305. Conducting a water balance analysis is an interesting idea that might be informative, but it has its own challenges because multiple sources of information need to be used to close the water budget, each of which has its own uncertainties. Furthermore, such an approach would only be useful for validating the ET data but would not provide information about the GPP and H performance. In view of these reasons, and because the other referee asked us to reduce the length of the manuscript, we have chosen not to include this analysis. Nevertheless, to provide an additional line of evidence investigating the WECANN quality, we have now added a new section 4.4 with an uncertainty analysis (which is much briefer than would be required for a full discussion of a water budget comparison). |
| Fourth, the spatial domain should be clearly defined. The authors said it is global product, but it did not include Antarctica and Greenland. | Thank you for the comments. We have now revised the description in the introduction section to clearly note what |

| | |
|---|---|
| Given the coarse resolution (100 km), most islands are likely uncovered but the global map (Fig 2) showed fluxes in some islands. How did it happen? Also, how to treat with water fraction for each 1-degree pixel? | the coverage of the new product is (Page 2, Lines 26-29), and provided a land mask in Figure S1. |
| Fifth, I recommend showing global uncertainty maps for GPP, LE, H. I think one of strengths in WECANN is its ability to quantify uncertainty. Show the uncertainty map and discuss where and why uncertainties are high. Also quantify uncertainties in global values (e.g. XXX PgC yr-1 +- Y PgC yr-1). | In the revised manuscript, we now include uncertainty estimates based on errors in the input data propagated into the network. We report a global average value as error is spatially and temporally variable. The new section 4.4 in the revised manuscript provides details on our uncertainty analysis and the results that are provided in Figure 14. |
| Sixth, test global more carefully. When I look at Fig 2, I found higher ET in mid to south east South America (e.g. cerrado) compared to other global ET products. Also, your ET in this region is relatively very high compared to your GPP map. So, water use efficiency will be very low in this region, which is unlikely. See global distribution of C4 maps. Higher proportion in C4 in this area is likely to lead higher water use efficiency. It is notable that your ANN did not consider C4 information. | The referee's point is an important one. The SIF relationship with GPP will likely change in C4 plants. However, we explicitly did not want to impose the C4/C3 (or even CAM) delimitation in the artificial neural network as it would be highly dependent on the quality of the classification map used. Given that we do not have partitioning of transpiration to total ET, it would be impossible to say whether the water use efficiency is indeed low or if rain re-evaporation and soil evaporation is the main process explaining the difference. We note that all training products include C3/C4 delimitation and therefore the C3/C4 delimitation is implicit in the training dataset and therefore can be learnt by the network. We have added a comment in the text emphasizing the referee's points (Page 9, Lines 17-19). |
| **Specific comments** | |
| P6: why only 21 FLUXNET sites were used? More than 150 sites data are open to public | We had selected these 21 sites to represent a range of climatic conditions along a geographical gradient for validation of our retrieval. Presenting evaluation metrics and temporal time series for 150 sites would lengthen the manuscript and make it now hard to read. However, in the revised manuscript we present summary statistics from a comparison of WECANN retrievals against a much larger number of tower data (97) from the FLUXNET 2015 and the La Thuile synthesis dataset in the supplementary tables S1-S3. We also comment on the results in Section 4.4. |
| P6 L23-24: The authors explained that target data is used for training, validation, and testing. I am confused with the terminology of validation and testing. How do they differ? Also, in L36, "after training, . . ... was evaluated". Here, does "evaluation" indicate validation or testing? I recommend clearly defining each term, and use them consistently across the whole manuscript. | We apologize for the confusion. The training, validation and testing proportions are related to the training phase of the retrieval. The back propagation algorithm uses a portion of the training data for training (basically estimating the weights of each neuron), and other portions of the training data for validation and testing that aims at checking the convergence of the training step. While after the training is done, we use a subset of data that were not used in the |

| | training process for evaluation. We revised the text in the new version of the manuscript to clarify these terminologies. (Page 7, Lines 5-10) |
|---|---|
| P6 L30: NN -> ANN (?) | Our apologies, this has been corrected in the revised manuscript. |
| P7 L9: Please define "multiple datasets." Is this training dataset?} | This refers to the three products that we use (together with error weights from Triple Collocation) to define a target dataset for training. We revised the text in the new version of the manuscript to clarify this. |
| P7 L12: What is "this" in "this prior distribution"? | It refers to the pseudo Bayesian training mentioned in the lines before. We revised the text in the new version and clarified the point. |
| P8 L20: Is this "target estimate" from 3.2? | Yes, this is the same. We made changes to section 3 and 4 of the manuscript in the new version to clarify all these terminologies. |
| P8 L22: Add another unit for GPP as PgC yr-1, which could be easily compared to the other studies. Same for LE (km3).} | Thanks for noting this. We included the new units along with previous ones in the new version of the manuscript. |
| P9 L29: I was surprised to see the reduction of GPP in the Saharan Desert after removing SiF. How to interpret this as we know there must be zero GPP? Also, exclusion of SiF in LE made mixed tendencies in this region. As we are confident LE and GPP are close to nil in this area, it will be interesting to test the impacts of inclusion/exclusion in SiF on LE and GPP here.} | This observation is true, and is caused by noise in the SIF data in deserts. As noted correctly by the referee, LE and GPP are close to zero in this region; therefore, the difference between the two retrievals (with and without SIF) divided by the small amount of flux in this region is on the order of the noise level in the retrievals. While the percentages of change are notable, the absolute values of difference between the two retrievals are less than 2 w m$^{-2}$ for LE and less than 0.7 gC m$^{-2}$ day$^{-1}$ for GPP. In addition, the noisy pattern does not show up in the H retrievals in this region. This is also another sign that the change patterns in LE and GPP are due to noise. However, due to the request of both referees we have revised our section on the impact of SIF, and our analysis now focuses on the differences between a retrieval with SIF or with NDIV/EVI (the new Section 4.5). Therefore, this figure was removed from the manuscript. |
| P10 L3: All three R2 looks too similar, so it is hard to tell 0.96 is higher than 0.94.} | Yes, we agree and have typically highlighted with bold fonts all comparable products for a fair comparison. This is further emphasized in the text of the revised version of the manuscript. |
| P10: The authors compared WECANN to FLUXNET-MTE, ECMWF, GLEAM and MODIS-GPP which were the training data for WECANN. I feel there should be self-correlation, so I am curious whether this is a reasonable approach. | The focus of this comparison is not validation. Since we used the three training products to generate the target dataset, we compare WECANN to these three to examine how similar is it to each of those training datasets. And we show that spatially WECANN is more similar to the product that has the lower RMSE in our TC estimates. |
| P10 L8: I know there are few eddy flux tower data in India, so FLUXNET-MTE might involve higher | It is true that there are few towers in India, but our retrieval does not rely solely on regional towers to estimate surface |

| | |
|---|---|
| uncertainty. However, this is the same situation for WECANN as it used FLUXNET-MTE and others, which are all uncertain as training dataset. | fluxes. Indeed this is a major advantage over FLUXNET-MTE and others. We train an artificial neural network algorithm using the three training products (two of which are not based on flux towers, and so do not necessary have higher uncertainty in certain areas because there are few towers there) mentioned in the manuscript for all the pixels However, the actual time-scale retrieval is mostly informed by the remote sensing observations (see discussion in Jimenez et al. 2009). That means we use the information from all the pixels over the globe to train one retrieval algorithm. This algorithm uses remote sensing observation at each point in time and space to retrieve surface fluxes. Therefore, lack of FLUXNET towers in any part of the globe would not impact the accuracy of WECANN retrievals, while this would expected to be more be an issue for products that upscale tower-based observations to estimate fluxes across the globe. |
| P10 L4: Be quantitative. Report bias. | Thanks for noting this. The point we have raised in this line (on the spread of scatter plots) can be quantitatively compared using the RMSD value that are provided in the figures. In the revised manuscript, we report this in the text. |
| P10 L20: Define "G" | Corrected in the revised manuscript. |
| P11: Many contents in this page should move to Methods. | In the revised manuscript, we re-organized the text and moved these contents to section 2.3. |
| P12 L5-6: Then why not removing this site given obvious deficiencies? | We believe that it is informative to include this comparison, as it illustrates that the representativeness area can be a challenge in comparing large-scale remote sensing based retrievals to point based tower data. In this case, we have good knowledge of the site and its surrounding region so it is possible to investigate if the tower data is representative of the larger scale fluxes. |
| P12- : As the authors well recognized, I feel it is odd to compare 1 degree WECANN to several hundred meters in flux towers. All discussion from this comparison seems too subjective. I think "validation" of 0.5-degree product is unlikely possible. As your products are too coarse, I would recommend evaluating at larger scales. For example, look interannual variability of global GPP (PgC yr-1), ET (W m-2), and H (W m-2) and compare to atmospheric inversion estimates. Test whether your product could capture big climate extreme events such as Russian heatwave, Texas drought etc. Compare to other existing global land surface products | While there is some caveat in validation against point based tower data, these are the only ground based observations available for such a validation. Moreover, in the comparison against tower data many large scale variabilities, including but not limited to the seasonal cycle are comparable to pixel based retrievals. This is also the case for interannual variability, and we have discussed them in detail, in section 4.4 of the original manuscript (section 4.3 of the revised manuscript). For instance the phenology has a strong impact on the seasonal cycle of the fluxes and is here clearly highlighted when comparing the different products to flux tower estimates. In the revised manuscript, we highlighted this limitation clearly in section 4.3, while noting that comparison against ground-based tower observations is common practice and |

| which were not used as input/training dataset in WECANN. | is what the community indeed looks for when a new retrieval algorithm is developed. We believe that specific drought or flood events would lack the generality provided here when comparing all years/months. Moreover, such a comparison needs detailed analysis that would further lengthen the manuscript (Indeed, the other referee asked us to reduce the length of the manuscript). In addition, in the new section 4.4 we provide uncertainty estimates of the retrievals along with interannual variability of surface fluxes at global scale to provide an additional line of evidence on the quality of the WECANN dataset. |

**Water, Energy, and Carbon with Artificial Neural Networks (WECANN): A statistically-based estimate of global surface turbulent fluxes using solar-induced fluorescence (Manuscript # bg-2016-495)**

| Comments | Responses/Actions |
|---|---|
| This manuscript is well written and deserves consideration for publication in this journal. However, I have the following issues that need to be addressed. | We appreciate the referee's positive feedback and provide responses to his/her comments below. |
| The paper proposes an empirical machine learning 'meta-model' to try to learn from different existing datasets to combine their strengths and factor out their limitations. On one hand, I appreciate this effort to bring together different datastreams and somehow harmonize them through this new consolidated product, but on the other, I am wary of this approach of blindly adding further algorithmic layers without really trying to understand mechanistically why the initial datasets have shortcomings. If all products are equally off in some parts, combining them just gives the false impression we are going in the right direction while reality is still off. Also, the FLUXNET-MTE used as training is already a machine learning product driven by various input variables, very much like WECANN is. Furthermore, there is quite some circularity in the work since the FLUXNET-MTE and MODIS GPP are both strongly based on the same fluxtowers used here for validation. I deem that all these points need to be acknowledged clearly and discussed thoroughly. | We acknowledge this concern, and would like to bring the following points to the referee's attention:
1- The WECANN machine learning retrieval is quite different from FLUXNET-MTE in the sense that we use remote sensing observations as inputs while FLUXNET-MTE upscales tower-based observations to estimate surface fluxes at global scale. Although both approaches use machine learning techniques (artificial neural networks in the case of WECANN and regression in the case of FLUXNET-MTE) their retrieval algorithms are quite different and directly informed by only remote sensing observations in WECANN, which we believe is an important means of better constraining the retrievals.
2- Moreover, our training approach uses all the spatial and temporal observations during the training period (2008-2010) to develop one single neural network for the global retrievals. This network is then used with remote sensing observations as input to retrieve surface fluxes. Therefore, if a few percentage of times and pixels, all the three training products are equally off this will be mitigated by the larger number of pixel/time data points that have more accurate estimates in other places and other times. In addition, the network can even correct the seasonal cycle when learning from an incorrect seasonal cycle training data, as the remote sensing inputs provide the information on the seasonal cycle directly. This has already been demonstrated previously, cfg. Jimenez et al. (2009).
Jimenez, C., Prigent, C., Aires, F. (2009). Toward an estimation of global land surface heat fluxes from |

multisatellite observations. Journal of Geophysical Research-Atmospheres, 114(D6), D06305.

3- On the issue of validation against FLUXNET tower data, we acknowledge that two of the training products use FLUXNET data for their calibration or as input. However, virtually all products have been calibrated in some ways or tested against eddy-flux tower, so implicit circularity is hard to avoid; there simply isn't another high quality data-set available. This does not however mean that the products are not independent: indeed the training products we use are typically calibrated to reproduce either the annual mean or are adjusted per season at very few sites (but not the exact temporal structures of the eddy-covariance observations except for FLUXNET-MTE). In addition the specific years of observations used here were not used in the calibration of MODIS and FLUXNET-MTE.

Here, we use the three training products together with a priori weights calculated from Triple Collocation to define a target dataset that has collective information from all three of them. And then we train our network on the target dataset. Finally, we validate the retrievals of WECANN against FLUXNET tower data and compare its performance with the performance of the three training products. While some information from FLUXNET observations propagates through the training products to WECANN training, the comparison results against FLUXNET observations show that WECANN learns from the three products collectively and performs better than any of them individually, emphasizing that our strategy works well. In addition, it is clear that WECANN does not have the seasonal biases seen in most retrievals (see e.g. FI-Hyy site where WECANN correctly captures this cold region's photosynthesis and evapotranspiration compared to the other products). Nevertheless, we have tried to also provide alternative lines of evidence to support the WECANN data quality, including an entirely new uncertainty analysis in section 4.5

We also made changes to sections 3 and 4 of the manuscript in the revised version to better reflect on these points. Moreover, in the new section 4.4 we now provide uncertainty estimates on WECANN retrievals to provide an additional line of evidence on the quality of the WECANN dataset.

| | |
|---|---|
| Could you specify why you use the SYN products (Level 3) from CERES instead of the EBAF ones (Level 3B)? The later have been energy balanced according to the product specifications. Wouldn't this be an advantage in your case? | Our goal here, as also mentioned in section 1 of the manuscript, is to only use remotely-sensed observations as input. The EBAF product is based on a model with some remote sensing observations; therefore, we decided to use the SYN product to avoid any model addition. |
| In the construction of the ANN, I would welcome to have some justification of why tangent sigmoid transfer functions are used instead of linear ones. I know this is often done, but it seems very arbitrary. | In this case, we tried the tangent sigmoid (the common choice) as well as linear, and did not see any notable changes in the performance of the network. Therefore, we chose to use the typical tangent sigmoid function. This has been added to the text (Page 6, Line 36 – Page7, Line 2). |
| Also, I did not quite understand how the 20% of 'testing' data is used. I clearly see that 60% are used for training and 20% for validation, but how exactly do you use the other 20%? Perhaps this just needs some rephrasing in the text for clarification. | We apologize for the confusion on this matter. In the revised manuscript, we explained this in more detail (Page 7, Lines 5-10). In summary, these percentages are for the data that are used in the training process. This is standard practice in artificial neural networks training. The back-propagation algorithm uses the training portion of the data for estimating the weights of the neuron in the network, and the validation and test data are used to evaluate convergence of the training. These are separate than the data that we used for validation later on. Our validation, uses a subset of data that are not used in the training, to make sure the network is not over-fitted to the training data. We revised the text in the new version of the manuscript and clarified the definitions. |
| Comparison with fluxtower measurements is not appropriate as the difference in spatial support is just too different (1 squared degree vs <1km2). Saying that WECANN performs better that other products based on individual towers while all these products cover such a larger area (by several orders of magnitude) just does not make much sense (even if it has been done in other studies). The authors would need to do some filtering of the towers to select only those that can be considered representative (e.g. http://doi.org/10.1016/j.rse.2016.04.027), although I doubt this would leave many valid towers for pixels of 1 squared degree. Another option that may be more feasible would be to make an evaluation at a larger aggregation scale, such as for clusters of similar climates and plant functional types. Making such averages from the fluxtowers on one side and from all pixels that are comparable in this respect on the other would reduce the number of measurements for validation, but would render them more credible. I would also suggest to exploit more of the | We acknowledge that comparison against point based tower data has its own limitation (as we also have noted in the manuscript), but these are the only ground based validation data that is available for evaluating a new global product. For this reason, we used a selection of sites spanning a geographical gradient and provided detail explanation on the comparison results in each site based on the knowledge of the land cover / land use around the site to make sure the differences, if any, can be explained either by uncertainties in WECANN retrieval or representativeness of the of towers. As the referee notes, filtering the towers based on representativeness might not leave us with any tower to use.
In the original manuscript, we only used 21 towers that were selected to represent a wide range of climatic conditions and we would be able to explain the results of each one of them, including the time series plots in detail. However, due to the request of both referees in the revised manuscript we include summary statistics from comparison of WECANN against 97 FLUXNET sites from three datasets: FLUXNET2015, La Thuile Synthesis Dataset and the Large-scale Biosphere-Atmosphere (LBA) experiment in Brazil. Results are provided in Tables S1 – S3 and discussed in Section 4.3 of the revised manuscript. We also want to |

| available towers in the Fluxnet2015 dataset instead of only 21. | emphasize that some features of the flux towers such as phenology, seasonality are correctly picked up by our retrieval compared to other products and are only moderately affected by the heterogeneity within the pixels (except if there would be a very different even composition of deciduous and conifers for instance). |
|---|---|
| The part pretending to demonstrate the value of SIF is also inadequate as the authors only test the effect of removing this one input. By doing so, any information of the actual vegetation phenology is lost, which would necessarily reduce the performance. What would be interesting would be to show that SIF provides better information that the classical vegetation indices like NDVI or EVI. To do so, the SIF input of the ANN should be replaced by one of these and then a judgement on the pertinence of SIF can be made. | We appreciate the referee's comment on this point. We have now included comparisons with retrievals that have only NDVI or EVI instead of SIF in the revised manuscript. This better shows the value of having SIF as an input in retrieving surface fluxes. Thank you for this important comment. The results further emphasize the difference between SIF and purely vegetation structure and phenology (as well as saturation effects of vegetation indices). Section 4.5 in the revised manuscript provides the detailed comparison of these retrievals. |
| Finally, the manuscript is often too long and too descriptive in several parts describing the graphs and maps. This needs to be reduced drastically. Most of what is being said can be easily inferred from the reader by looking at the graphs, while deeper discussion on why discrepancies occur between products and fluxtowers would be more welcome. Also, please remove the extensive references to different parts of the text and the description of the structure of the paper (e.g. page 3 lines 10-20), I think they are lengthening the text needlessly. | Given the novelty of the approach we feel that it is important to correctly describe the different steps of the analysis as many are relatively new such as the machine learning and the triple collocation. We had received the opposite comments before that we were not sufficiently describing the details; hence, the reason why the article goes into the details of the retrievals. We have edited he manuscript throughout and shortened it where possible. |

[revised manuscript text omitted]

---

## Author Response (AR2)

Response to Referee 1

**Water, Energy, and Carbon with Artificial Neural Networks (WECANN): A statistically-based estimate of global surface turbulent fluxes using solar-induced fluorescence (Manuscript # bg-2016-495)**

| Comments | Responses/Actions |
|---|---|
| Thank you for the efforts in trying to address the questions raised during the review process. I think the manuscript has been improved, specifically with the inclusion of the NDVI/EVI tests, the larger fluxnet dataset for validation and the analysis on the uncertainty estimation. However, I am still not satisfied on some issues. In particular on two points that were raised by both reviewers: (1) the circularity of using flux data in both the training and the validation and (2) the mismatch between the footprint of the towers and the 1 degree cells of the products. For both points, I would strongly encourage including the analyses suggested by Reviewer 1, that is: for point 1 to do an evaluation of ET at basin scale from water balance; and for point 2 to focus on an evaluation at larger scale such as catching big climate events, and perhaps the inter-annual variability of GPP and comparing them perhaps with atmospheric inversion estimates. I believe that these will be much more convincing in showing the added value of WECANN that the current individual pixel to tower comparisons/validations that are done. These could be placed in supplementary material in my opinion, if the length is really an issue. | We thank the referee for their comments and suggestions. In order to address these critiques, we have implemented the following three validation analyses:

1- Evaluating WECANN estimates in three extreme heatwave events: Russia 2010, Texas 2011, US corn belt 2013 (Section 4.4 in the new manuscript).

2- Estimating the error of WECANN ET estimates in five large basins (Amazon, Colorado, Congo, Mississippi, and Orinoco) using data from a water budget closure model (Section 4.5 in the new manuscript).

3- Calculating correlation between GRACE estimates of terrestrial water storage changes and P-ET from WECANN (Section 4.6 in the new manuscript).

Results of these analyses show the capability of WECANN retrievals in capturing large-scale anomalies. |
| I do realize that both suggestions of Reviewer 1 where countered in part by using the argument that I (reviewer 2) asked to reduce the length of the manuscript. I am afraid that I was misunderstood on this point. I did not mean the whole manuscript should be reduced per se, but rather that some specific parts were unnecessarily lengthy. Large parts describing the plots and maps could (and still can in my opinion) be reduced, mostly the 'Results and Discussion' section. Also, phrases like "In this section, we [do | We changed the text throughout the manuscript to reduce the unnecessary sentences as needed. |

| | |
|---|---|
| this]… " after the subtitle saying precisely what 'this' is just adds unnecessary length and could be reformulated in my opinion e.g. top of page 9 and page 10. Bottom-line is that tackling the two points raised above should have a priority over reducing the length of the paper as a whole. | |
| Regarding the other arguments for keeping to a 'standard' comparison of 1 degree vs fluxtower footprints, such as the this is how it is often done or that fluxtowers is the only source of validation GPP, I believe now would be the time to raise the issue more prominently that doing these matches is sub optimal as we are unable to disentangle the signal from the confounding factors caused by intra-pixel spatial heterogeneity. The current manuscript would benefit from taking that point up front and show a more innovative 'validation' using the suggestions of reviewer 1. And I repeat, the current work on individual towers should be kept, as it is informative, but perhaps placed in supplementary if space is lacking. | We agree with the referee on the shortcomings of the FLUXNET tower comparisons given the coarse spatial resolution of WECANN retrievals. We have summarized the results of this comparison in a shorter section (4.2) now with only 5 flux towers being discussed in details instead of 17, and put more focus on the large scale validation efforts that are introduced in the new version. We have also summarized the results of flux tower comparisons in the new Figure 6 based on Plant Functional Types. |

**Other remarks/suggestions**

| | |
|---|---|
| The comparison against a set of 97 fluxsite is welcome, but this should come to the forefront, instead of being relegated to supplementary material. Not with the full details of all individual fluxtower comparison (which could stay in the supplementary material), but with a synthetic table with the average values for each flux. In this table also include the average values for the RSME and Bias. You could also consider using a symmetric index of agreement (https://www.nature.com/articles/srep19401) as a generic value from 0 to 1 to specify the degree of agreement. This combines together the effects of RMSE, bias and correlation in a single metric, which also does not assume a reference. Indeed, here it could be argued that the fluxtower estimate is not an absolute reference because it is not measured on the same observational support as the 1 degree WECANN estimate. | As noted in the previous comment, we have summarized the flux tower comparison section and brought it up into the first evaluation section (4.2) following presenting the seasonal patterns from WECANN retrievals. We have added the average RMSE of each product in Tables S1-S6. But we tend to not report the average bias, because it can be misleading. For example, if a product has large negative and positive biases across different sites a mean value close to zero it does not mean that it has low bias compared to another product that has only small positive biases that on average are larger than the mean of the first product. |
| The point on C3/C4 plants raised by reviewer 1 is not explicitly addressed in the text. I would expect a phrase talking about it somewhere. | We added text to Page 9 Lines 20-23 in the current manuscript to explain this further. |
| Page 2, Line 31: you would need to specify that you " jointly retrieve surface turbulent fluxes" for | Thanks for the note on this. We have revised the text accordingly. |

| | |
|---|---|
| the first time using SIF, as GPP has been retrieved from SIF on many occasions | |
| Uncertainty analysis is very welcome. Clarify in the caption of figure 14 that these are the annual mean estimates of each flux with their uncertainty. As it is now, it seems the centre of the box plot represent the mean uncertainty instead of the mean flux. | We changed the figure caption to clarify the data presented in this figure. |
| You say the decreasing trends in GPP and LE are consistent in figure 14. They are consistent with each other (with is to be expected from your product) but is it consistent with the general knowledge of what is really happening across the globe (a marked reduction of GPP since 2011) ? Please put this in context with other studies. Also, how do you explain the strange pattern of sudden decrease in 2007-2009 for GPP in the Mid latitudes? | While we believe analysis of the trends in the retrievals is beyond the scope of this manuscript (and it is the focus of our next research study being carried out), here we present a figure to compare the yearly anomalies of LE, H, and GPP across the globe and different latitudinal zones and compare them with FLUXNET-MTE (Figure R1). Since FLUXNET-MTE is only available until 2011, we can't compare the anomalies from 2012-2015. However, comparison of the results from 2007-2011 shows consistent anomalies and trends in both datasets. |
| In figure S1, there seems to be a strange asymmetric effect of your land mask, e.g. there is a white line on the west coast of Africa and South America, but not on the East coast. Are the datasets properly aligned? | Thanks for noting this issue. This was caused by an error in using the plot function in MATLAB. It is corrected in the new version. |

[Figure]

Figure R1- Annual mean anomalies of LE (top row), H (middle row) and GPP (bottom row) retrievals at global (left column) and regional (four right columns) scales for WECANN (solid lines) and FLUXNET-MTE (dashed lines).

Response to Referee 2

**Water, Energy, and Carbon with Artificial Neural Networks (WECANN): A statistically-based estimate of global surface turbulent fluxes using solar-induced fluorescence (Manuscript # bg-2016-495)**

| Comments | Responses/Actions |
|---|---|
| Editor comments.

Failing to find a suitable reviewer in time, I've made my own assessment. I largely agree with the other reviewer, but would like to point our that in addition I find this manuscript to be weak in discussing the implication of the findings in context to other studies. The reviewer and my comments may give a direction how to improve this, in general I think what is needed is a criticial assessment of what one can learn from this development, which we did not know in the past, and how for instance the trend estimates WECANN produces compare to other studies (or whether they maybe should not be regarded as trends given the brevity of the record.) One fairly simple addon would be to plot the other products as well in Figure 14, to have a more synthetic overview. | We thank the editor for his comments and suggestions. We have addressed each of the comments in the following and implemented necessary changes in the manuscript to address them.

One of the key findings of this study is showing the capability of SIF measurements to be used for estimating latent and sensible heat fluxes along with GPP. GPP retrievals from SIF have been studied before, but our retrieval is the first of its kind to estimates latent and sensible heat.

This finding is of interest to a large community of researchers in our field, in particular science teams of OCO-2, OCO-3, ECOSTRESS, geoCARB, and FLEX missions. During the last couple of months we have been invited to their science team meetings and presented our results on this topic which was highly received by their team members. |
| I would suggest to keep some (say 4 site from different biomes) of the FLUXNET comparison plots in the main manuscript (and in agreement to the other reviewer, but them more at the front of the results), and put the others into the Appendix | We have reduced the flux tower comparison section by including only 6 sites, and summarizing the statistics from Tables S1-S3 in Figure 6 based on Plant Functional Types. |
| **Minor comments** | |
| P1 L29: GPP and NPP are not generally considered turbulent exchanges, nor are they a flux from the land to the atmosphere. The turbulent flux observed is the net ecosystem exchange rate, of which GPP is a compartment flux. I also don't see the reason to include NPP in this list if it is not subject of this study. | We removed NPP from the discussion. |
| P1 L34 redundant sentence | We revised this in the new version. |
| P2 L1: this is not only the MPI-BGC, see Tramontana, G. et al. Predicting carbon dioxide | We revised the text here to appropriately refer to this product. |

| | |
|---|---|
| and energy fluxes across global FLUXNET sites with regression algorithms. Biogeosciences 13, 4291-4313 (2016). | |
| P2 L9ff: To be fair to the regression approach discussed above, this paragraph needs to mention that any model product that is used to train an ANN necessarily inherits all its imperfections in capturing means, extremes and so on to the ANN. | We already have mentioned this further down in the same paragraph, Page 2 Lines 14-16 in the new version. |
| P2 L20: It is necessary here to introduce the training data sources (at least in an overview). | Introduction on training data is added. (this is now in Page 3, Lines 7-8). |
| P2 L30: its is not clear what the first key innovation is at this point. Also SIF has been introduced already. Please edits the text such that the information flow is more logical, e.g. by mentioning SIF first here. | We revised the content in the first section to appropriately address this issue. |
| P3 L9: True, but how does this work in an ANN, which does not intrinsically know about this coupling? | The way ANN develops such a coupling implicitly is by using one set of weights and biases for its neurons to estimate the three fluxes together. These weights and biases are tuned during the training. Therefore, having a good target dataset for training is essential to develop an accurate network. |
| P3 L14-18 can be safely removed. | We have removed this section. |
| P3 L 22 why 1° when all input data are at a higher resolution? | As listed in Table 2, three of the input datasets (Air Temperature, Net Radiation, and Precipitation) are posted on 1° spatial grids. Therefore, the final product can't be on a higher spatial resolution. |
| P6 L8ff. Please clarify which data has been used, and why. It is probably best to include a supplementary table, listing all sites, data coverage and their sources. | As we have explained later in this section, we are using data accessed through the FLUXNET 2015 dataset. This is a product developed by the PIs of all flux towers contributing data, and all the specifications of data are specified in the product itself. So it would be redundant to list them again here. Based on the request of PIS, we have appropriately cited the data, acknowledged them in the acknowledgement section and used the correct abbreviation for site names. |
| P7 L20. To show that the method adds information, would it be helpful to compare the product against the point-wise average of the three input data sources? | As our results show WECANN has the best performance among the four products considered in this study. WECANN is developed by characterizing the errors of each of the training products and learning from them collectively while minimizing errors. Moreover, as our TC analysis shows error of each of the training products change from one pixel to another and there is no single product that has the lowest/highest error at all pixels. Therefore, a simple averaging of the pixel-wise estimates will not provide a reasonable estimate of the fluxes at all pixels and times. |

| | |
|---|---|
| P8 L13. Help the reader what the error means. Does the method simply fail, or is this evidence for the inconsistency between the different data sets. If so, what is the knowledge gain relative to simply looking at the difference between the three products? | Here, 'error' refers to the variance of the random error component of each measurement system estimated by TC technique. This information is an indicator of the uncertainty in each of the measurement systems with respect to the truth and can't be achieved by looking at the differences between the measurements themselves. We have clarified this in the new version. |
| P10 L4: Is this really surprising, given that MET and WECANN are both purely statistical models, whereas GLEAM and ECMWF contain process knowledge, which may obscure any direct relationship between fluxes and driving variables? | We believe this similarity between WECANN and FLUXNET-MTE is caused because of their accuracy. FLUXNET-MTE is driven by point based measurements from flux towers unlike GLEAM and ECMWF which are physical models driven by other environmental variables. Therefore, FLUXNET-MTE is expected to be more representative of the true fluxes. On the other hand, WECANN learns from all three products by minimizing the information from the most uncertain product in each case and it is driven by independent satellite based observations. This results in WECANN to have the best performance, and indirectly be more similar to FLUXNET-MTE. |
| P15 L18. I must have missed this, which figure/statistics demonstrates the added value of SIF at the global scale? Section 4.5 only discusses site-level analyses | This is based on the site level comparisons as we have discussed in Section 4.8. |
| P15 L 21: This is in parts not surprising, because WECANN isn't fully independent from these data. This needs to be mentioned here. | We tend to disagree with the editor in this case. Because FLUXNET-MTE (which is the other product used in training and our validation comparisons) is a direct upscaling of the flux tower data, while WECANN is indirectly trained on FLUXNET-MTE as well as three other products which are independent of flux tower data. Therefore, we would have expected that FLUXNET-MTE, which is not independent of flux tower data, be the best product. But this is not the case, and WECANN which learns from three independent products during training and uses SIF as an input has a better performance. |
| Figure 9ff: There seems to be an issue with some of the fluxnet data in the year 2014-2015. Please check and potentially remove flawed data. | There was no panel f in Figure 9 in the previous version of the manuscript. We are not sure what the editor is referring to. But we have investigated flux tower data as much as possible before including them in the analysis. In cases that the reported values seems unrealistic we discussed some of them in the manuscript. |
| Figure S1: Why is the land-mask shifted relative to the continents? | Thanks for noting this issue. This was caused by an error in using the plot function in MATLAB. It is corrected in the new version. |
| Tables S1-S4: Why is only R2 averaged over the sites, and not the others? I think it would be helpful to provide the average and standard deviation across sites of these statistics as a table | We had only averaged R2 because it is the index with similar range for all cases, unlike RMSE and bias that their values should be interpreted by considering the mean of the flux at each site. |

| | |
|---|---|
| for the manuscript (while keeping the SI tables), to make this material more accessible to the reader. | However, due to the request of the other reviewer as well, we have now added the average for RMSE. But we tend to not report the average bias, because it can be misleading. For example if a product has large negative and positive biases across different sites a mean value close to zero it does not mean that it has low bias compared to another product that has only small positive biases that on average are larger than the mean of the first product.
Moreover, the new Figure 6 summarizes the mean and one standard deviation from these tables based on Plant Functional Types. |

[revised manuscript text omitted]

---

## Author Response (AR3)

Response to Editor

**Water, Energy, and Carbon with Artificial Neural Networks (WECANN): A statistically-based estimate of global surface turbulent fluxes using solar-induced fluorescence (Manuscript # bg-2016-495)**

| Comments | Responses/Actions |
|---|---|
| Dear authors,

many thanks for your revised manuscript, which have generally improved the quality of the manuscript. After reading the revised version I regret to inform you that there is a lot of editorial matters (and one important scientific issue), which you will need to address point-by-point, including a statement on how you have changed the manuscript, before your manuscript can be published in Biogeosciences.

Best wishes,
Sönke Zaehle | Dear Dr. Zaehle,

We appreciate your comments. In the following we have provided point-by-point response and actions taken to address your comments.
We have also attached a track-changed version of the manuscript for your consideration.

We look forward to hearing back from you soon.

Regards,
S. Hamed Alemohammad |
| The GRACE analysis is not helpful as it currently is. a) Is the PDF of the correlation coefficient correct? The visual impression of the map suggests much more positive and negative correlation than near-zero correlation, whereas the PDF suggests that the correlation is centred around zero nearly in a bell-shaped form with few values above 0.4. The latter suggests to me a rather random agreement with no model skill. b) You cannot blame snow melt on the lacking correlation between GRACE and WECANN, as snow melt does not change the water storage. c) Snowmelt also does not explain the main other regions (typically with low LE), which have inverse correlation to GRACE. | The PDF of the GRACE correlations was/is correct. Note that the colorbar for near-zero values and for no-value pixels where both white. We changed the colorbar so that near-zero values be represented with gray color.

The goal of the GRACE analysis was to provide an additional independent evaluation of the ET estimates from WECANN (as requested by the reviewers) at global scale. We now agree with the editor that the GRACE analysis does not meet this goal – there are simply too many simplifications in the correlative analysis (e.g. lack of accounting for runoff, imperfections in storage and precipitation data, low number of samples to calculate discrete time steps and correlations, etc…) that added noise, leading the PDF to be centered near zero. To show that the poor results of the GRACE analysis were due to errors in the analysis (and assumptions) and not issues with WECANN, we note that repeating the analysis using FLUXNET-MTE data vs. WECANN data over the same temporal domain does not appreciably degrade or improve the results (see Figure RR1 at the end of this response). Therefore, we have removed the GRACE analysis from the text for clarity. |

| | We believe the other independent evaluations (extreme events and basin scale ET) provide satisfactory evaluations of the WECANN retrievals beside the comparisons with eddy covariance tower data. |
|---|---|
| There is no such thing as model validation. Please use the term evaluation or similar instead. | We changed the terminology throughout the manuscript from validation to evaluation when interpreting the results. Please note that we did retain the use of the word validation in the paragraph describing cross-validation of the ANN data as it refers to specific machine learning terminology. |
| Sorry for being pedantic, but GPP is the gross primary production of the vegetation and cannot be observed directly. GPP contributes to the net exchange of $CO_2$ between ecosystems and the atmospheric boundary layer, of which the overwhelmingly turbulent part is observed as net ecosystem exchange. It is incorrect to refer to GPP as a turbulent flux, as it is also not correct to insinuate that GPP was directly measured by eddy covariance (P2 L4). GPP is derived from NEE measurements using a flux partitioning algorithm. Please revised all instances in the text and title to clarify the issue. | We agree with the editor on the definition of GPP and how it is estimated from NEE at eddy covariance flux towers. We revised the text in P2 L4 to clarify this. GPP is often times referred to as a flux, for example in:

 Reichstein, M., Stoy, P.C., Desai, A.R., Lasslop, G., Richardson, A.D., 2012. Partitioning of Net Fluxes, in: Aubinet, M., Vesala, T., Papale, D. (Eds.), Eddy Covariance. Springer Netherlands, Dordrecht, pp. 263–289. doi:10.1007/978-94-007-2351-1_9

 Ma, J., Yan, X., Dong, W., Chou, J., 2015. Gross primary production of global forest ecosystems has been overestimated. Sci. Rep. 5, 10820. doi:10.1038/srep10820

 We nonetheless corrected the terminology throughout the paper accordingly. And the new title reads: "Water, Energy, and Carbon with Artificial Neural Networks (WECANN): A statistically-based estimate of global surface turbulent fluxes and gross primary productivity using solar-induced fluorescence" |
| Abstract
 The abstract is not written very effectively. E.g. sentence two and three each mention three data sets / sources, but it isn't clear whether they are the same, or different and what these are. Similar, somewhat later you state that the retrievals are evaluated against FLUXNET observations, but then a sentence later refer to these as in situ eddy covariance measurements. Again, it is unclear whether that's the same or not, and why this needs to be mentioned twice. The abstract also misses that other studies in the past have used SIF, the point here is that it's (probably) the first JOINT retrieval of H, LE and GPP. In L21, please be more specific about what WECANN produces. | We revised the abstract in the new version of the manuscript to remove repetitions and clarify the datasets, the scope and novelty of the work. We clarify that this is the first *joint* global estimate of H, LE and GPP using SIF.

 The reference to the El Nino impact is based on the results in Section 4.7 that we discuss the global patterns in 2015/2016 and their relationship to the El Nino event. This has now been clarified. |

| | |
|---|---|
| It is not clear to me what "is constrained by SIF information" means. Of course WECANN is constrained by SIF because its used in the training - be more specific about what SIF contributes to the study.

The last sentence does not make a lot of sense. The uncertainty analysis has nothing to do with the extreme event analysis. I also disagree with the ENSO link, which is not demonstrated in either the extreme event analysis nor in Figure 14 (too few El Ninos in there to make a case). | |
| Introduction
For me, the flow of the introduction would be more easy to follow if the sentence on p2 L 17 was move to after P3 L1 (new paragraph after "in addition to GPP), because then all the information about your novelties are in one place. To help the reader, you could also add in p 2 L8, "Alternatively, as done here, one can…". | Thanks for the suggestion. We revised the Introduction accordingly. |
| P3 L1 "Using our machine learning technique…" This sentence comes a bit out of nowhere: in the previous sentence, you talk very generally about the potential of SIF, but this sentence now talks about something that will be the outcome of your study. For me, this would be more fitting to the conclusion section than here. | We moved this statement to the conclusion. |
| Data
Add reference to Table 1/2 to first view sentences of the Section "2 Data" at the respective places. (then nobody wonders where to find the reason for the 1° resolution.) | References are added in the revised manuscript. |
| Use of FLUXNET data. It is not redundant to clarify the stations you've used, because you apply a filter on the number of stations used (clearly not all 750 stations are used, assuming that not 653 are coastal stations), and add (without providing any motivation or reasoning) other data-bases.
It is important to clarify, which station has been used, and for which years, and why data from the LBA and La Thuille dataset have been used.
It is important to document, which stations and data have been used. The supplementary table is a good start, if you added the exact site-years used. | In the previous version of the manuscript, all supplementary tables S1 – S6 had the full list of sites used with the name of the network that the data are retrieved from. We have now added the years/months used to these tables. |

| | |
|---|---|
| In the description of the FLUXENT evaluation data, it's worth mentioning that WECANN and the data are not fully independent, because the FLUXNET-MTE used for the training relies on these data. This does not preclude you from using these data to evaluate your product, but it needs to be mentioned somewhere. | The editor is correct. We added text in Section 4.2 on this topic in the revised manuscript (P10, L17-20). |
| For consistency reasons, I would move the description of the ET estimates based on runoff also into the Data section. | Revised in the new manuscript. |
| Method
I don't see how you've addressed my comment on previous version P3 L9 in the revised manuscript. For me this is a point that merits discussion in the discussion section of the ms. E.g. P7 L19 is a place at which it is worthwhile pointing out that no assumptions are made that H LE and GPP are in anyway connected, i.e. any correlation between them entirely relies on the training data sets. | We use one neural network to retrieve Le, H and GPP together. Therefore, the coupling of these three is learned by the network implicitly during training. We added the following text to P7 L11-14 to address this topic in the manuscript:
"During the training, the network implicitly learns the coupling of the LE, H and GPP by using one set of neurons (with their respective weights and biases) to estimate the three variables. This is an advantage of using a machine learning technique that eliminates the need to define physical relationships between different variables." |
| Results
Supplementary tables: you should provide average statistics of every metric used for the FLUXNET evaluation (such as the bias), as it is meaningful to know whether the a product on average is higher or lower than the other estimates. Of course, one single metric does not provide a full insight into whether a product is better or worse than another. | In the revised supplementary materials, we have added the average bias statistics across all sites for each product in Tables S1 – S6. |
| P12 L 24: this analysis is not at the global scale. | We revised the text to read: "In order to further assess WECANN at regional scales,…" |
| Section 4.5: As said above, the first few lines should be presentation of the data set in Section 2. It is unclear to me whether or not the difference is an absolute number, or whether the outcome is that WECANN is always larger than the other estimate? | We moved the introduction of water budget closure model to the new section 2.3.2 in the revised manuscript.
The differences are absolute differences. We changed the text to clarify this. |
| Given the focus of this manuscript on SIF, it would make a stronger case in Section 4.8 if Figure 6 & 7 also contained the WECANN-NDVI and WECANN-EVI. | We appreciate your comment on this but prefer not to include those results in Figures 6 & 7 for three reasons:
- Detailed results of WECANN NDVI and EVI are compared with WECANN in the supplementary tables. These also contain average values which provide similar comparison as Figure 6.
- If we include these in Figures 6 & 7 the flow of the paper would be broken as these two alternative |

| | retrievals with NDVI and EVI are introduced later on in section 4.8. |
|---|---|
| | - Figures 6 & 7 already have a lot of information, adding two more datasets to them would make them hard to read. |
| Figure R1 is much more informative about the model and its anomalies in time than Figure 14. You can keep the bar plots for WECAN, but it would be much more informative if you plotted the FLUXNET-MTE alongside with it. The text associated with this Figure should briefly state, how much (if any) information on the inter annual variation in WECAN is derived from FLUXNET-MTE. | We agree that this figure has valuable information. While we appreciate the suggestion to add Fluxnet-MTE (and other training data) to Figure 14 to mirror figure R-1, these data do not affect the uncertainty analysis, so that adding them would confuse the flow of the manuscript. Instead, we have included the figure in the new version of the supplementary material (Figure S7). We have also included anomalies from all three training products (for 2007-2011 that we have the data). We note that plotting anomalies for both WECANN and training datasets is a better comparison than their absolute values. Text is added at the end of Section 4.3 (P12 L31-38) to explain this figure.

 The inter-annual variability of WECANN is not derived by any of its training datasets, including FLUXNET-MTE. Indeed, the remote sensing observations that are input to the ANN are driving the inter-annual variability of the WECANN retrievals. We added text at the end of Section 4.6 to clarify this point. |

[revised manuscript text omitted]